# DreamDPO: Aligning Text-to-3D Generation with Human Preferences via Direct Preference Optimization

Zhenglin Zhou [1 2]  Xiaobo Xia [3]  Fan Ma [2]  Hehe Fan [2]  Yi Yang [1 2]  Tat-Seng Chua [3]

## Abstract

Text-to-3D generation automates 3D content creation from textual descriptions, which offers transformative potential across various fields. However, existing methods often struggle to align generated content with human preferences, limiting their applicability and flexibility. To address these limitations, in this paper, we propose DreamDPO, an optimization-based framework that integrates human preferences into the 3D generation process, through direct preference optimization. Practically, DreamDPO first constructs pairwise examples, then compare their alignment with human preferences using reward or large multimodal models, and lastly optimizes the 3D representation with a preference-driven loss function. By leveraging pairwise comparison to reflect preferences, DreamDPO reduces reliance on precise pointwise quality evaluations while enabling fine-grained controllability through preference-guided optimization. Experiments demonstrate that DreamDPO achieves competitive results, and provides higher-quality and more controllable 3D content compared to existing methods. Code is publicly available at: https://github.com/ZhenglinZhou/DreamDPO.

## 1. Introduction

3D content generation is pivotal in driving innovation across diverse fields, including product design, medical imaging, scientific visualization, and the rapidly growing domains of virtual and augmented reality (Li et al., 2023a). Despite its extensive applications, it remains challenging to create high-quality 3D content, which requires substantial time and effort, even for professionals. In response, *text-to-3D generation* has emerged as a solution by automating 3D generation from textual descriptions, which archives remarkable advancements in the field (Poole et al., 2022; Wang et al., 2022; Yu et al., 2023; Wang et al., 2024; Shi et al., 2023; Katzir et al., 2023; Chung et al., 2024; Wu et al., 2024b). Nevertheless, some researchers (Xie et al., 2024; Ye et al., 2025) emphasize that 3D content generated by existing methods often fails to align with *human preferences* fully, highlighting the need for continued refinement and innovation in these methods.

Previous work (Ye et al., 2025) has leveraged *reward models* to integrate human preferences into the generation process, leading to enhanced 3D generation outcomes. The core idea is to regularize the generated 3D content to achieve a high *pointwise score* from the reward model. Despite these improved results, several issues remain to be addressed. First, it heavily depends on the reward model's ability to accurately evaluate the *pointwise quality* of generated content, which places significant demands on the reward model. Second, since the reward model can only provide *quality-relevant* scores, it lacks the flexibility to enable controllability from other perspectives. The issues reduce the applicability and adaptability of the current method. This falls short of meeting diverse requirements or providing broader control in 3D generation, which is never our desideratum.

To relieve the issues of prior work and better align 3D generation with human preferences, we propose DreamDPO. Essentially, DreamDPO is an optimization-based method for text-to-3D generation. It achieves alignment through direct preference optimization, leveraging preferences derived from either *reward models* or *large multimodal models*. Specifically, DreamDPO operates by initializing a 3D representation (Mildenhall et al., 2020; Kerbl et al., 2023) and optimizing it through a three-step iterative process. First, *pairwise examples* are constructed *on-the-fly* by applying different Gaussian noise. Second, a reward model or a large multimodal model *ranks* these examples based on their matching with the input text prompt or *specific instructions*, which matches human preferences about the pairwise

---

[1]State Key Laboratory of Brain-machine Intelligence, Zhejiang University, China [2]ReLER, CCAI, Zhejiang University, China [3]National University of Singapore, Singapore. Correspondence to: Yi Yang <yangyics@zju.edu.cn>, Xiaobo Xia <xbx@nus.edu.sg>.

*Proceedings of the $42^{nd}$ International Conference on Machine Learning*, Vancouver, Canada. PMLR 267, 2025. Copyright 2025 by the author(s).

examples[1]. Finally, a reward loss is computed from the *pairwise preferences*, which guides the update of the 3D representation. By incorporating human preferences into the optimization loop, DreamDPO generates 3D assets that achieve superior alignment with textual inputs, along with enhanced texture and geometry quality.

DreamDPO can be justified as follows. While absolute quality evaluation inherently provides a ranking on pairwise examples, a ranking requires only the scores to distinguish *relative preferences*, but need not be perfectly accurate (*c.f.*, (Zhang et al., 2024d)). DreamDPO takes advantage of this distinction, and changes the previous *score-guided* optimization (Ye et al., 2025) to *preference-guided* optimization. Therefore, it lowers the demand for precise scoring and requires only distinguishable scores. Additionally, DreamDPO can make use of the preferences provided by large multimodal models. By constructing preferred and less preferred examples based on specific instructions about the attributes of generation content (*e.g.*, the object number and motion), it directs the optimization process to align more closely with the desired outcomes. This strategy enhances adherence to instructions and introduces *fine-grained controllability*, meeting diverse requirements effectively. Moreover, we conduct a series of experiments to justify our claims. Empirical results demonstrate that our method flexibly accommodates either reward models or large multimodal models, which enables the generation of higher-quality and more controllable 3D content.

Before delving into details, we clearly emphasize our contribution as follows.

- Conceptually, we propose DreamDPO that shifts the paradigm of text-to-3D generation by reducing dependence on precise absolute quality evaluation. Instead, it leverages distinguishable relative preferences and integrates large multimodal models, which achieves remarkable alignment with human preferences while enabling fine-grained controllability.

- Technically, DreamDPO pioneers a three-step optimization process, combining online pair construction, preference ranking, and ranking-driven updates. This innovative design ensures superior alignment with input prompts, significantly enhancing the quality and adaptability of generated 3D content.

- Empirically, extensive results establish DreamDPO as a new benchmark for text-to-3D generation. Both quantitative and qualitative results are thoroughly analyzed, supported by detailed discussions and ablation studies.

---

[1]Reward models are trained using pairwise data reflecting human preferences, while large multimodal models are inherently aligned with these preferences (Sun et al., 2023).

DreamDPO outperforms 13 state-of-the-art methods, achieving the best quantitative performance across two key metrics while delivering highly impressive and controllable qualitative results.

The rest of this paper is organized as follows. § 2 introduces some background knowledge relevant to this work. § 3 presents the technical details of our proposed DreamDPO. Experimental results are analyzed and discussed in § 4. Conclusions are given in § 5.

## 2. Preliminaries

Text-to-3D generation aims to create high-quality 3D assets aligned with a given text prompt $y$. The pipeline typically distills knowledge from a parametrized diffusion model $\epsilon_\phi$ (Rombach et al., 2022a; Shi et al., 2023) into a learnable 3D representation with parameters $\theta \in \Theta$ (*e.g.*, NeRF (Mildenhall et al., 2020), DMTet (Shen et al., 2021), and 3DGS (Kerbl et al., 2023)), where $\Theta$ is the space of $\theta$ with the Euclidean metric. Score distillation sampling (SDS) (Poole et al., 2022) is used to guide the distillation process.

**Diffusion models.** The diffusion model has been widely used in generative tasks (Sohl-Dickstein et al., 2015; Song et al., 2022; Bao et al., 2022; Peebles & Xie, 2023). Generally, it involves a forward process to gradually add Gaussian noise to data points and a reverse process to transform Gaussian noise into data points from a target distribution $p_{\text{data}}$. The reverse process starts from an initial noise $\mathbf{z}_T \sim \mathcal{N}(\mathbf{0}, \mathbf{I})$. At each diffusion step $t$, the model refines noisy data $\mathbf{z}_t$ into a cleaner one $\mathbf{z}_{t-1}$ until finally producing $\mathbf{z}_0 = \mathbf{x} \sim p_{\text{data}}$. Therefore, the transitions $p(\mathbf{z}_{t-1}|\mathbf{z}_t)$ can be learned effectively by the diffusion model.

**Score distillation sampling (SDS).** SDS was proposed in DreamFusion (Poole et al., 2022), and has been widely studied (Wang et al., 2024; Yu et al., 2023; Katzir et al., 2023; Chung et al., 2024; Wu et al., 2024b; Zhuo et al., 2024; Ye et al., 2025). Technically, for a rendered image $\mathbf{x}$ from a 3D representation, random noise $\epsilon$ is added at timestep $t$. A pre-trained diffusion model predicts this noise. The SDS loss is computed as the difference between predicted and added noise, which optimizes a set of parameters $\theta$. The gradient of the SDS loss with respect to $\theta$ is:

$$\nabla_\theta \mathcal{L}_{\text{SDS}} = \mathbb{E}_{t,\epsilon}[w(t)(\epsilon_\phi^s(\mathbf{x}_t; y, t) - \epsilon)\frac{\partial \mathbf{x}}{\partial \theta}], \quad (1)$$

where $\mathbf{x}_t = \alpha_t \mathbf{x}_0 + \sigma_t \epsilon$, $w(t)$ is a weighting function, and $s$ is a pre-defined scalar of classifier-free guidance (CFG) (Ho & Salimans, 2022). The minimization of the SDS loss follows the score function of the diffusion model to move $\mathbf{x}$ to the text description region, ensuring the generated 3D representation aligns with the given text prompt. Note that to make a continuous and smooth presentation, we provide

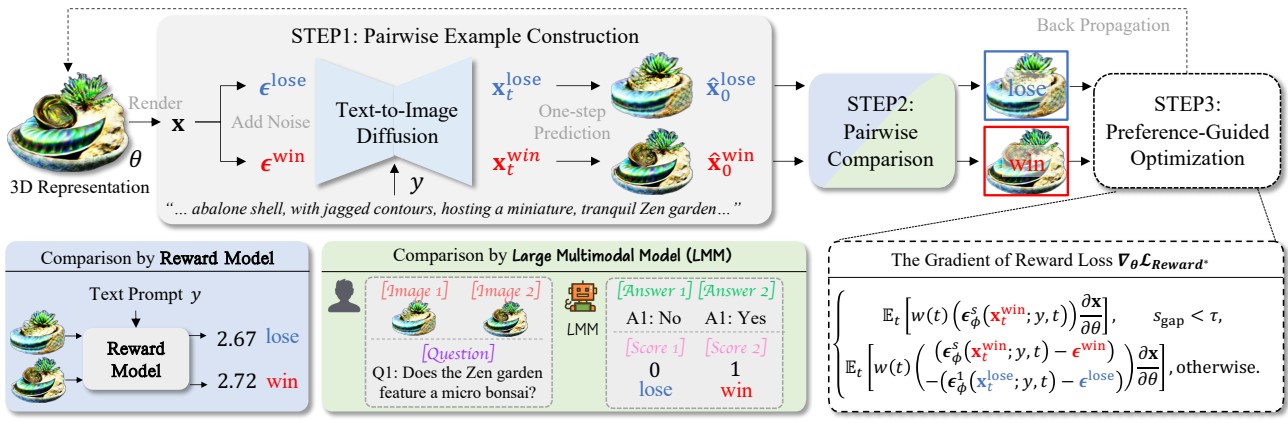

*Figure 1.* Overview of our method. DreamDPO first constructs pairwise examples, then compares their alignment with human preferences using reward or large multimodal models, and lastly optimizes the 3D presentation with a preference-driven loss function. The loss function pulls the `win` example $\mathbf{x}_t^{\text{win}}$ closer and pushes the `lose` example $\mathbf{x}_t^{\text{lose}}$ away. As a piecewise objective, it selectively pushes $\mathbf{x}_t^{\text{lose}}$ only when the preference score gap $s_{\text{gap}}$ exceeds a threshold $\tau$, preventing chaotic gradients from overly similar $\mathbf{x}_t^{\text{lose}}$.

a detailed review of related work in Appendix A.

## 3. Method

**Overview.** DreamDPO is an optimization-based text-to-3D generation method. It begins by initializing a 3D representation, *e.g.*, NeRF (Mildenhall et al., 2020). In each training iteration, the optimization procedure involves three key steps: (1) *pairwise example construction*: pairwise examples are generated online by applying different Gaussian noises during the diffusion process; (2) *pairwise example comparison*: a reward model or a large multimodal model (LMM) compares the generated examples based on their alignment with the desired text prompt; and (3) *preference-guided optimization*: a piecewise reward loss is calculated using the pairwise comparison, and the 3D representation is updated accordingly. Overall, our DreamDPO guides the optimization process with human preferences, leading to 3D assets with improved alignment to input text and enhanced texture/geometry quality. The framework overview of our method is provided in Figure 1. A complete algorithm flow of our method can be checked in Appendix B.

### 3.1. Algorithm Details

**Pairwise example construction.** Given a sampled camera pose, an RGB image $\mathbf{x}$ can be rendered from the 3D representation using renderers. Then two different Gaussian noise $\boldsymbol{\epsilon}^1$ and $\boldsymbol{\epsilon}^2$ are added to $\mathbf{x}$ at timestep $t \in [0, T]$, resulting in pairwise noisy images $\mathbf{x}_t^1$ and $\mathbf{x}_t^2$:

$$\mathbf{x}_t^1 = \alpha_t \mathbf{x}_0 + \sigma_t \boldsymbol{\epsilon}^1, \quad \mathbf{x}_t^2 = \alpha_t \mathbf{x}_0 + \sigma_t \boldsymbol{\epsilon}^2, \quad (2)$$

where $\mathbf{x}_0 = \mathbf{x}$, $\alpha_t$ and $\sigma_t$ are hyperparameters satisfying $\alpha_0 \approx 1, \sigma_0 \approx 0, \alpha_T \approx 0, \sigma_T \approx 1$ (*c.f.*, (Sohl-Dickstein et al., 2015; Ho et al., 2020)). Afterward, we feed the pairwise noisy images into a pre-trained text-to-image diffusion

model $\boldsymbol{\epsilon}_\phi$ (Shi et al., 2023; Rombach et al., 2022a) and generate corresponding predictions:

$$\hat{\mathbf{x}}_0^1 = \frac{\mathbf{x}_t^1 - \sigma_t \boldsymbol{\epsilon}_\phi(\mathbf{x}_t^1; y, t)}{\alpha_t},$$
$$\hat{\mathbf{x}}_0^2 = \frac{\mathbf{x}_t^2 - \sigma_t \boldsymbol{\epsilon}_\phi(\mathbf{x}_t^2; y, t)}{\alpha_t}, \quad (3)$$

where $\hat{\mathbf{x}}_0^1$ and $\hat{\mathbf{x}}_0^2$ are predicted $\mathbf{x}_0$ of a single step for $\mathbf{x}_t^1$ and $\mathbf{x}_t^2$, respectively (Song et al., 2021).

**Pairwise comparison.** After pair construction, at step $t$, we utilize a rank model denoted by $r(\cdot)$ to compare $\mathbf{x}_t^1$ and $\mathbf{x}_t^2$. This yields a preferred prediction $\mathbf{x}_t^{\text{win}}$ and a less preferred one $\mathbf{x}_t^{\text{lose}}$, where $\mathbf{x}_t^{\text{win}} = \mathbf{x}_t^1$ and $\mathbf{x}_t^{\text{lose}} = \mathbf{x}_t^2$, or vice versa. It is worth noting that our DreamDPO supports both reward models (Xu et al., 2024b; Wu et al., 2023) and LMM-based AI annotators (Bai et al., 2023; Yang et al., 2023), where reward models are used as default.

**Preference-guided optimization.** The proposed method leverages the pairwise comparison $(\mathbf{x}_t^{\text{win}}, \mathbf{x}_t^{\text{lose}})$ to enable efficient sampling via optimization to yield human preferred 3D assets. To achieve this, we need a differentiable loss function, where preferred images have low losses and less preferred images have high losses. To this end, inspired by (Rafailov et al., 2024; Meng et al., 2024; Wallace et al., 2024), we reformulate SimPO (Meng et al., 2024) to eliminate the need for a reference model and derive a differentiable objective:

$$\mathcal{L}_{\text{Reward}} = -\mathbb{E}_t \big[ w(t) \left( \| \boldsymbol{\epsilon}^{\text{win}} - \boldsymbol{\epsilon}_\phi(\mathbf{x}_t^{\text{win}}; y, t) \|_2^2 \right. \\ \left. - \| \boldsymbol{\epsilon}^{\text{lose}} - \boldsymbol{\epsilon}_\phi(\mathbf{x}_t^{\text{lose}}; y, t) \|_2^2 \right) \big], \quad (4)$$

where $\boldsymbol{\epsilon}^{\text{win}}$ and $\boldsymbol{\epsilon}^{\text{lose}}$ denote Gaussian noise for $\mathbf{x}_t^{\text{win}}$ and $\mathbf{x}_t^{\text{lose}}$ respectively. Intuitively, $\mathcal{L}_{\text{Reward}}$ encourages $\boldsymbol{\epsilon}_\phi$ to pull $\mathbf{x}_t^{\text{win}}$ closer and push $\mathbf{x}_t^{\text{lose}}$ further away.

*Table 1.* Qualitative comparisons on 110 prompts generated by GPTEval3D (Wu et al., 2024a). We calculate the ImageReward score (IR) (Xu et al., 2024b) for human preference evaluation, the CLIP score (Radford et al., 2021) for text-image alignment evaluation, and GPTEval3D (Wu et al., 2024a) for comprehensive 3D quality evaluation. The best performance in each case is shown in bold.

| Method | IR ↑ | GPTEval3D ↑ | | | | | |
|---|---|---|---|---|---|---|---|
| | | Alignment | Plausibility | T-G Coherency. | Geo Details | Tex Details | Overall |
| DreamFusion (Poole et al., 2022) | -1.51 | 1000.0 | 1000.0 | 1000.0 | 1000.0 | 1000.0 | 1000.0 |
| DreamGaussian (Tang et al., 2023) | -1.56 | 1100.6 | 953.6 | 1158.6 | 1126.2 | 1130.8 | 951.4 |
| Fantasia3D (Chen et al., 2023) | -1.40 | 1067.9 | 891.9 | 1006.0 | 1109.3 | 1027.5 | 933.5 |
| Instant3D (Li et al., 2023c) | -0.91 | 1200.0 | 1087.6 | 1152.7 | 1152.0 | 1181.3 | 1097.8 |
| Latent-NeRF (Metzer et al., 2023) | -0.42 | 1222.3 | 1144.8 | 1156.7 | 1180.5 | 1160.8 | 1178.7 |
| Magic3D (Lin et al., 2023) | -1.11 | 1152.3 | 1000.8 | 1084.4 | 1178.1 | 1084.6 | 961.7 |
| Point-E (Nichol et al., 2022) | -2.24 | 725.2 | 689.8 | 688.6 | 715.7 | 745.5 | 618.9 |
| ProlificDreamer (Wang et al., 2024) | -0.50 | 1261.8 | 1058.7 | 1152.0 | 1246.4 | 1180.6 | 1012.5 |
| Shap-E (Jun & Nichol, 2023) | -2.10 | 842.8 | 842.4 | 846.0 | 784.4 | 862.9 | 843.8 |
| SJC (Wang et al., 2023a) | -0.82 | 1130.2 | 995.1 | 1033.5 | 1079.9 | 1042.5 | 993.8 |
| SyncDreamer (Liu et al., 2023c) | -1.77 | 1041.2 | 968.8 | 1083.1 | 1064.2 | 1045.7 | 963.5 |
| Wonder3D (Long et al., 2024) | -1.70 | 985.9 | 941.4 | 931.8 | 973.1 | 967.8 | 970.9 |
| MVDream (Shi et al., 2023) | -0.58 | 1270.5 | 1147.5 | 1250.6 | 1324.9 | 1255.5 | 1097.7 |
| DreamDPO (ours) | **-0.35** | **1298.9** | **1171.9** | **1276.4** | **1373.2** | **1296.9** | **1203.1** |

Following (Poole et al., 2022), we consider the gradient of $\mathcal{L}_{\text{Reward}}$ and omit the U-Net Jacobian term for effective optimization. It leads to the following gradient for optimizing 3D representations with preference pairwise comparisons:

$$\nabla_\theta \mathcal{L}_{\text{Reward}} = \mathbb{E}_t \left[ w(t) \left( \boldsymbol{\epsilon}_\phi(\mathbf{x}_t^{\text{win}}; y, t) - \boldsymbol{\epsilon}^{\text{win}} \right) - \left( \boldsymbol{\epsilon}_\phi(\mathbf{x}_t^{\text{lose}}; y, t) - \boldsymbol{\epsilon}^{\text{lose}} \right) \frac{\partial \mathbf{x}}{\partial \theta} \right]. \quad (5)$$

However, in practice, the gradient in Equation (5) fails to produce realistic results (refer to Figure 6). Delving into the optimization process, we observe that the pairwise comparison results can be overly similar, leading to nearly equal scores. In this case, directly pushing $\mathbf{x}_t^{\text{lose}}$ away leads to chaotic gradients. To address this, we introduce a piecewise optimization loss that selectively pulls $\mathbf{x}_t^{\text{win}}$ when the preference score gap $s_{\text{gap}}$ is small. The gradient of the final loss is defined as:

$$\nabla_\theta \mathcal{L}_{\text{Reward}*} := \begin{cases} \mathbb{E}_t \left[ w(t) \left( \boldsymbol{\epsilon}_\phi^s(\mathbf{x}_t^{\text{win}}; y, t) \right) \frac{\partial \mathbf{x}}{\partial \theta} \right], & s_{\text{gap}} < \tau, \\ \mathbb{E}_t \left[ w(t) \left( \boldsymbol{\Delta}_t^{\text{win}} - \boldsymbol{\Delta}_t^{\text{lose}} \right) \frac{\partial \mathbf{x}}{\partial \theta} \right], & \text{otherwise.} \end{cases} \quad (6)$$

where $\tau = 0.001$ is a pre-defined threshold, $\boldsymbol{\Delta}_t^{\text{win}} := \boldsymbol{\epsilon}_\phi^s(\mathbf{x}_t^{\text{win}}; y, t) - \boldsymbol{\epsilon}^{\text{win}}$, and $\boldsymbol{\Delta}_t^{\text{lose}} := \boldsymbol{\epsilon}_\phi^1(\mathbf{x}_t^{\text{lose}}; y, t) - \boldsymbol{\epsilon}^{\text{lose}}$. Note that $s_{\text{gap}} := r(\mathbf{x}_t^{\text{win}}, y) - r(\mathbf{x}_t^{\text{lose}}, y)$ indicates the discrepancy of preference scores between $\mathbf{x}_t^{\text{win}}$ and $\mathbf{x}_t^{\text{lose}}$. Instruction questions can also be incorporated into LMM-based ranking models to provide explicit guidance (see empirical evaluations in §4.3).

## 4. Experiments

In this section, a series of experiments are conducted to justify our claims. We first detail experiment setups (§4.1). The comprehensive results and comparison with previous advanced methods are then presented and discussed (§4.2). Finally, we carry out an analysis study to further elaborate on and discuss the superiority of our method (§4.3).

### 4.1. Experimental Setups

**Datasets and measurements.** We here evaluate the proposed method with 110 prompts from GPTEval3D (Wu et al., 2024a), which covers a range of creativity and complexity use cases. Based on this, two evaluation strategies are exploited. (1) We utilize a text-to-image reward model named ImageReward (Xu et al., 2024b) to evaluate human preference for 3D assets. We calculate the average preference score across 120 rendered images of a 3D asset and its corresponding text prompt. (2) We use GPT-4V to perform pairwise comparisons with baselines, generating Elo ratings that align with human judgments on text alignment, 3D plausibility, and texture-geometry coherence, *etc*. More details of the two measurements can be found in Appendix C.1.

**Baselines.** Following GPTEval3D (Wu et al., 2024a), we benchmark our method against 13 baselines categorized into text-guided and image-guided approaches respectively. Specifically, the text-guided group includes DreamFusion (Poole et al., 2022), DreamGaussian (Tang et al., 2023), Instant3D (Li et al., 2023c), Fantasia3D (Chen et al., 2023), Latent-NeRF (Metzer et al., 2023), Magic3D (Lin et al., 2023), MVDream (Shi et al., 2023), Point-E (Nichol et al., 2022), ProlificDreamer (Wang et al., 2024), Shap-E (Jun & Nichol, 2023), and SJC (Wang et al., 2023a). Besides, the image-guided group includes SyncDreamer (Liu et al., 2023c) and Wonder3D (Long et al., 2024).

**Implementation.** We conduct experiments using Py-

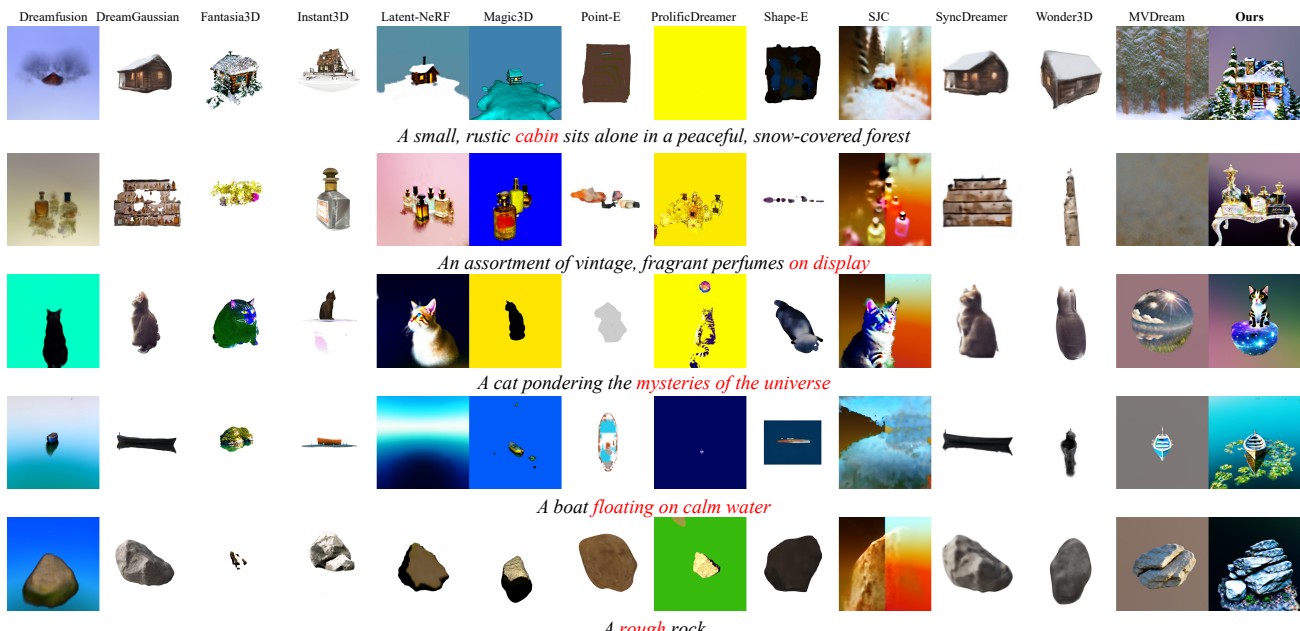

Figure 2. Qualitative comparisons on the benchmark of GPTEval3D (Wu et al., 2024a). Existing methods struggle with text matching, as marked in red. DreamDPO improves text matching, which provides better human preference results. (Zoom in to see the details.)

Torch (Paszke et al., 2019) and threestudio (Guo et al., 2023), with MVDream (Shi et al., 2023) as the backbone of our method. Note that we use PyTorch auto-differentiation to compute analytic normals for geometry evaluation in GPTEval3D and do not use the Lambertian shading trick (Lin et al., 2023) due to memory limitation. We follow the training strategy of MVDream and use HPSv2 (Wu et al., 2023) as the default reward model. The optimization process takes around two hours on a single NVIDIA RTX A6000 GPU.

### 4.2. Comparison with Prior Methods

#### 4.2.1. QUALITATIVE COMPARISONS

We conduct three qualitative evaluations, which include comparing 13 benchmarks of GPTEval3D, MVDream (Shi et al., 2023), and DreamReward (Ye et al., 2025). The evaluations show improvements in text alignment, generation stability, and texture-geometry details, respectively. As seen in Figure 2, while the comparing baselines produce high-fidelity results, they often fail in text alignment, as marked in red. For instance, in the prompt "*A small, rustic cabin sits alone in a peaceful, snow-covered forest*", most existing methods miss key elements like the forest (the first row in Figure 2). In contrast, our method accurately captures both objects, showcasing its effectiveness for improving text alignment. Further comparisons with MVDream, are shown in Figure 3. Although MVDream is capable of generating multiview consistent 3D assets, it struggles with long prompts (*e.g.*, the second and fourth rows in Figure 3). Instead, our method performs well across both short to long prompts. Lastly, the comparisons with DreamReward

demonstrate that our method not only improves text alignment (*e.g.*, ensuring "*leaves a trail of flowers*" appears under the bicycle shown in the first row in Figure 7) but also enhances geometric and texture details (*e.g.*, generating a more luxuriant oak shown in the second row in Figure 7). More visualization results can be found in Appendix D.2.

#### 4.2.2. QUANTITATIVE COMPARISONS

We provide extensive quantitative comparison results to justify our claims. Human preference evaluations are first conducted using ImageReward, as illustrated in the first column of Table 1. Our method achieves competitive performance compared to existing methods, highlighting the benefits of preference-guided optimization via our reward loss. Then we perform a comprehensive evaluation using GPTEval3D. The results indicate that our method outperforms previous state-of-the-art (SOTA) methods, and ranks first across all metrics. Specifically, our method achieves improvements in text-asset alignment (+28.4), 3D plausibility (+24.4), text-geometry alignment (+25.8), texture details (+48.4), geometry details (+41.4), and overall performance (+24.4). It showcases the superiority of the proposed method in enhancing text and geometry details while maintaining 3D consistency.

### 4.3. More Analyses and Justifications

**Evaluation on different backbones.** We here further investigate the impact of backbone selection on our method. The performance of DreamDPO using Stable Diffusion v2.1 (SD2.1) (Rombach et al., 2022a) is provided. Note

MVDream                    **DreamDPO (Ours)**

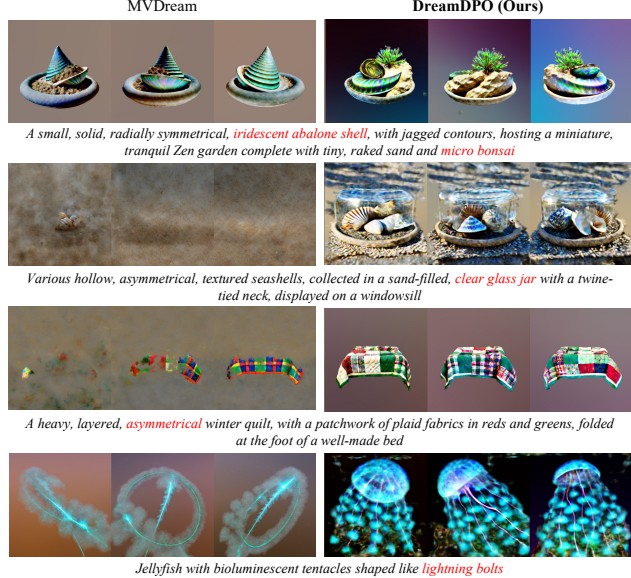

*A small, solid, radially symmetrical, iridescent abalone shell, with jagged contours, hosting a miniature, tranquil Zen garden complete with tiny, raked sand and micro bonsai*

*Various hollow, asymmetrical, textured seashells, collected in a sand-filled, clear glass jar with a twine-tied neck, displayed on a windowsill*

*A heavy, layered, asymmetrical winter quilt, with a patchwork of plaid fabrics in reds and greens, folded at the foot of a well-made bed*

*Jellyfish with bioluminescent tentacles shaped like lightning bolts*

Figure 3. Qualitative comparisons with MVDream (Shi et al., 2023). DreamDPO performs well across short to long prompts, offering better human preference results, marked in red. (Zoom in to see the details.)

ProlificDreamer    Ours (w. SD2.1)    MVDream    Ours (w. MVDream)

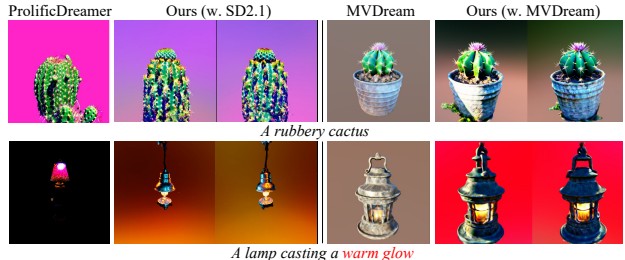

*A rubbery cactus*

*A lamp casting a warm glow*

Figure 4. The analysis of backbone. We present the results of DreamDPO using Stable Diffusion v2.1 (SD2.1) (Rombach et al., 2022a). DreamDPO demonstrates effective performance with SD2.1, highlighting its potential to leverage more advanced backbone diffusion models for further improvements.

that as the previous method ProlificDreamer (Wang et al., 2024) also utilizes SD2.1, we compare DreamDPO with SD2.1 against ProlificDreamer. As shown in Figure 4, our method can perform effectively with SD2.1 and achieve competitive results compared to ProlificDreamer. Importantly, DreamDPO does not need LoRA training and is *more efficient* than ProlificDreamer.

**Evaluation on different reward models.** We study the impact of reward model selection on our method. Specifically, ImageReward (Xu et al., 2024b) is used, which is an image-based reward model with a similarity function comparable to HPSv2 (Wu et al., 2023). As shown in Figure 5, the results demonstrate that our method performs effectively across different reward models, demonstrating its flexibility and scalability. For instance, in the prompt "*A mug filled with steaming coffee*", both HPSv2 and ImageRe-

MVDream      Ours (w. HPSv2)    Ours (w. ImageReward)

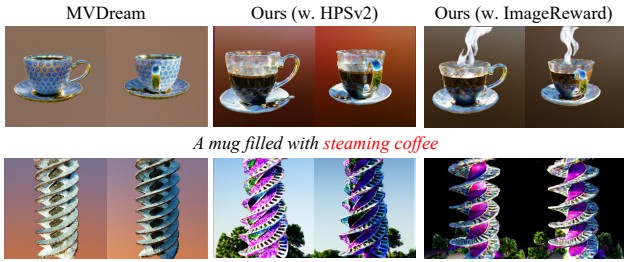

*A mug filled with steaming coffee*

*A twisted tower*

Figure 5. The analysis of reward models. We present the results of DreamDPO using ImageReward (Xu et al., 2024b). DreamDPO demonstrates effective performance with ImageReward, highlighting its potential to leverage stronger reward models to further enhance generation quality.

SDS      $\tau = 0.01$    $\tau = 0.005$    $\tau = 0.002$    $\tau = 0$

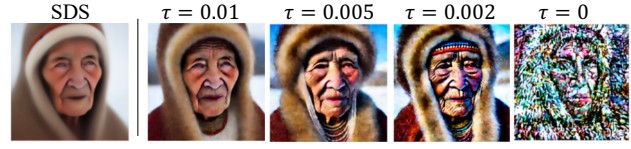

*An elderly nomadic woman with a deeply lined face, wearing a fur-lined cloak and a traditional woven headdress adorned with beads.*

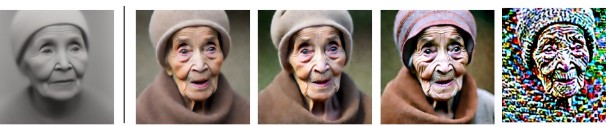

*A close-up of an elderly woman with deep-set wrinkles around her eyes, high cheekbones, and thin lips, wearing a knitted hat with intricate patterns.*

Figure 6. The analysis of the score gap threshold $\tau$. We conduct 2D toy experiments with $\tau$ ranging from 0.01 to 0. The results indicate that a small but non-zero $\tau$ effectively filters out overly similar lose examples, leading to more detailed outputs.

ward successfully capture the coffee, and ImageReward places greater emphasis on the steam. While ImageReward demonstrates improvement over the baseline, HPSv2 yields superior results due to its better generalization across diverse image distributions (Wu et al., 2023). Therefore, we adopt HPSv2 as our default reward model. It highlights the potential of leveraging stronger reward models, *e.g.*, VisionReward (Xu et al., 2024a), to further enhance generation quality. Furthermore, DreamDPO is compatible with rule-based reward metrics, such as image quality metrics (*e.g.*, BRISQUE (Mittal et al., 2011)). It does not require additional reward models and is suitable for scenarios with limited computational resources. As shown in Figure 10, we include a case study demonstrating DreamDPO with BRISQUE, a no-reference image quality evaluator, to show its effectiveness.

**Evaluation on different score gaps.** We investigate the impact of the score gap $\tau$. Specifically, 2D toy experiments with $\tau$ range from 0.01 to 0 are conducted. We provide results in Figure 6, which show that a smaller $\tau$ produces more detailed outputs. Note that a small $\tau$ means choosing to lose examples with scores close to win samples, and

MVDream DreamReward **DreamDPO (Ours)**

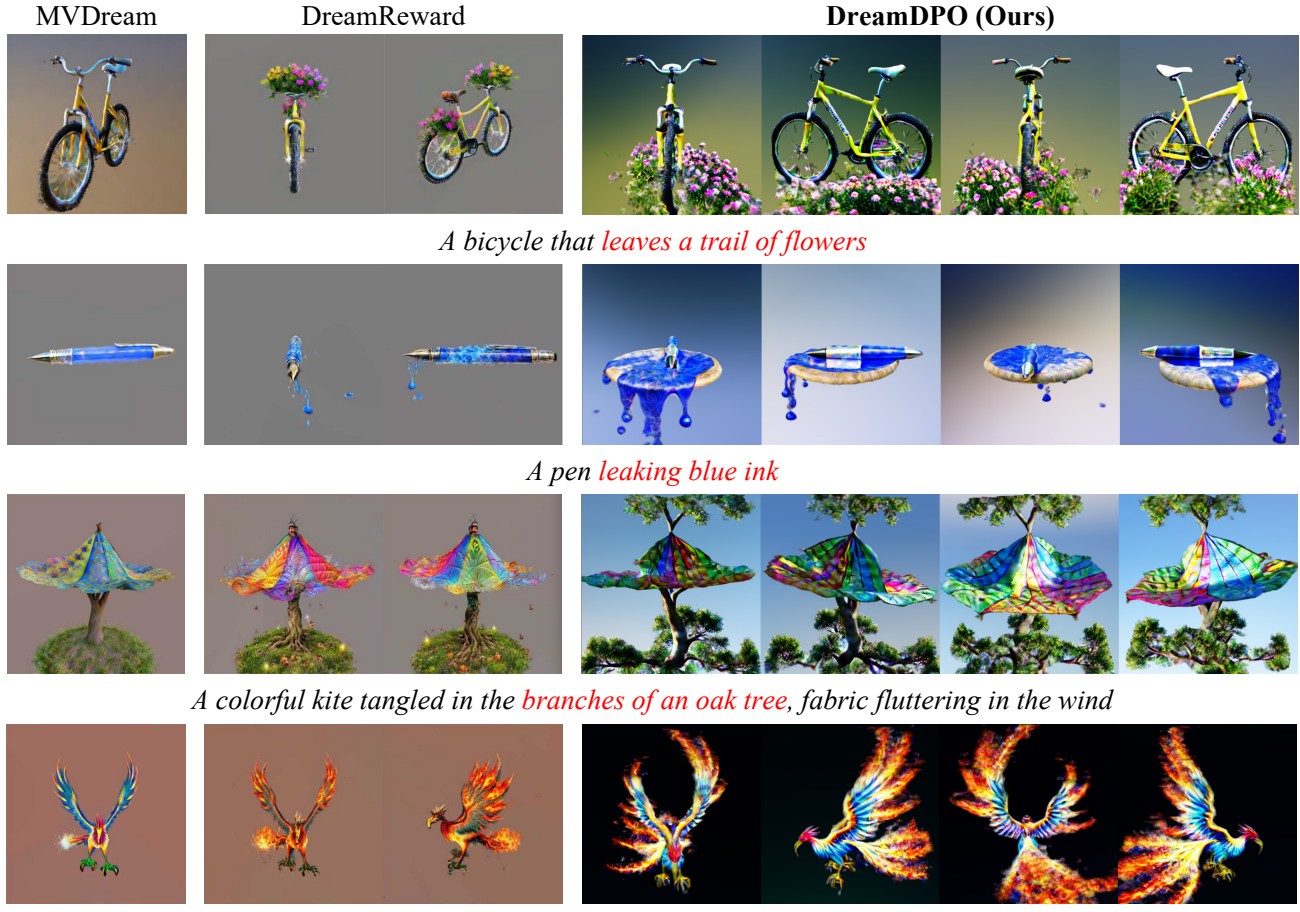

*A bicycle that leaves a trail of flowers*

*A pen leaking blue ink*

*A colorful kite tangled in the branches of an oak tree, fabric fluttering in the wind*

*An ultra-detailed illustration of a mythical Phoenix, rising from ashes, vibrant feathers in a fiery Palette*

*Figure 7.* Qualitative comparisons with DreamReward (Ye et al., 2025). DreamDPO improves both text matching (marked in red) and geometric/texture details.

focusing the training process on hard cases. However, we observe that $\tau = 0$ (the last column in Figure 4) results in a chaotic gradient. To balance high-fidelity generation and stable training, we suggest using *a small but non-zero $\tau$*, which excludes overly similar lose examples.

**Evaluation on different pair examples.** The influence of different pair example generation methods is studied. Specifically, we compare: (1) *different noises*, by adding different Gaussian noises with the same timesteps; (2) *difference steps*, by adding the same Gaussian noise with different timesteps. As shown in Figure 9, using different Gaussian noise yields better results than different timesteps. We attribute that noisy latents with different timesteps are easier to distinguish, making them less effective as challenging examples. It highlights the importance of generating meaningful and challenging lose examples. Accordingly, we adopt different noises as the default setting.

**Ranking model design.** We explore the potential of our method to leverage large multi-modal models (LMMs) for explicit guidance. Instead of relying on a reward model, we use large visual-language models, such as QwenVL (Bai et al., 2023), to rank paired results. Specifically, we extract "yes" or "no" questions from the text prompt, query the LMM with rendered paired examples, and count the "yes" responses to calculate the reward score. Then, we use Equation (6) with threshold $\tau = 1$ for 3D assets generation. As shown in Figure 8, our method effectively improves text alignment by using LMM to guide the optimization with user instruction (*e.g.*, correcting the *number* and *attribute* of 3D assets). Additionally, our method flexibly supports using lose examples to guide optimization. For example, given the prompt "*A dancing elephant*", the baseline generates an elephant standing rather than dancing. By setting "*elephant stays on the ground*" as the lose example and pushing it away, our method encourages the elephant to lift its leg, leading to a dancing pose. It highlights the potential of our method to integrate LMMs into 3D generation. More implementation details of the LMMs-based comparison can be found in Appendix B.2.

**Discussion on potential bias from reward models.** The

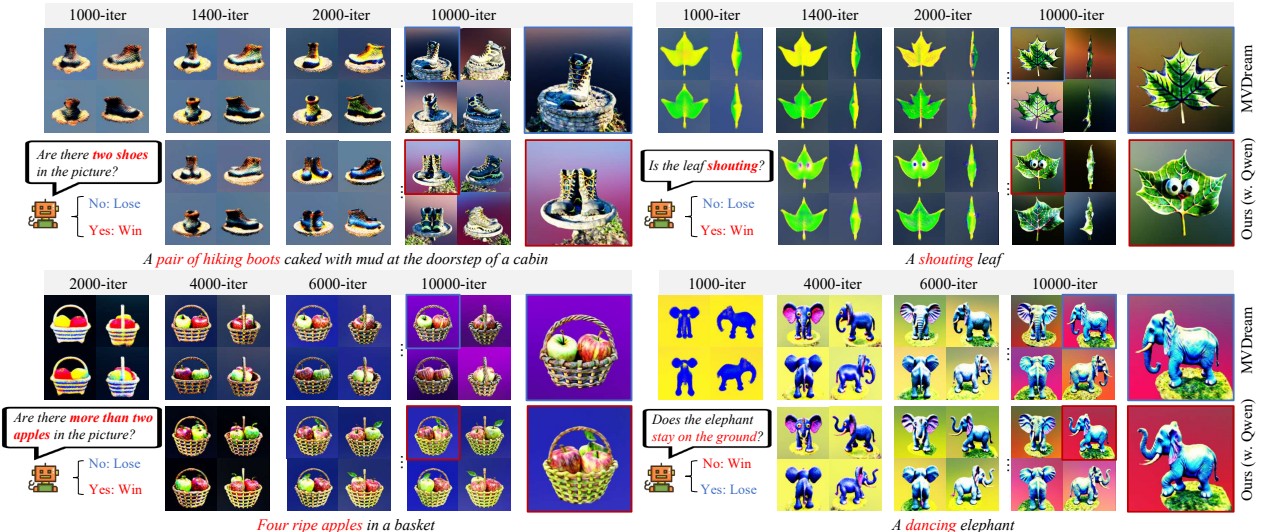

Figure 8. The generation results of DreamDPO with large multi-modal models (LMMs). We explore the potential of our method to leverage LMMs, such as QwenVL (Bai et al., 2023) for explicit guidance in correcting the number and attribute of 3D assets. The left corner shows the details of pairwise comparisons using the LMM, including the question and win/lose criteria. By carefully designing the question, DreamDPO can leverage both `win` and `lose` examples to guide optimization. (Zoom in to see the details.)

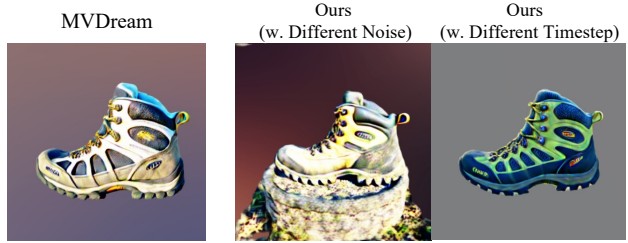

Figure 9. The analysis of pairwise example construction. We compare (1) different noises: adding different Gaussian noises with the same timesteps, and (2) difference timesteps: adding the same Gaussian noise with different timesteps.

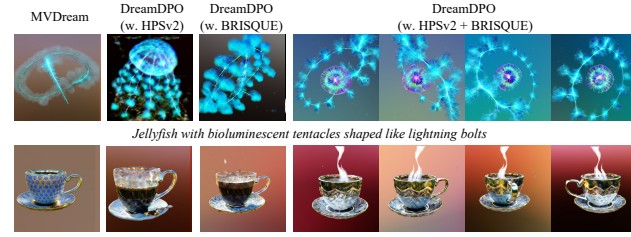

Figure 10. The analysis of rule-based reward metrics and ensemble reward models. (1) DreamDPO is compatible with image quality metrics (e.g., BRISQUE (Mittal et al., 2011)). (2) DreamDPO could combine two rankings to reduce reliance on a single model and mitigate unwanted biases.

reward bias is a known issue in RL-based optimization. One solution is to scale the training data and model size of the reward model. As shown in our experiments in Section 4.3, the reward model HPSv2, which has a stronger generalization ability compared to ImageReward, demonstrates superior generation performance in most cases. Moreover, DreamDPO supports an ensemble of reward models, combining their rankings to reduce reliance on a single model and mitigate unwanted biases (see Figure 10).

**Discussion on potential limitations on 2D metrics.** In this paper, we mainly adopt 2D reward models, due to the fact that they are relatively mature, with abundant data and good reward performance. However, they have limitations. While they improve prompt alignment, such as correcting character attributes (e.g., changing "man" to "knight"), they struggle to address inherent Janus issues (e.g., characters with three legs). Our DreamDPO is flexible with both 2D and 3D rewards (Ye et al., 2025). Experiments in Figure 11 demonstrate that it effectively mitigates Janus issues and

some potential artifacts, offering a more reliable evaluation for 3D consistency.

**Discussion on LMM-based reward model enhancements.** We investigate strategies to enhance performance when using large multimodal models (LMMs) as reward models, including increasing the number of comparison candidates and employing stronger LMMs. First, we extend DreamDPO to support multi-sample comparisons. Specifically, we modify STEP1 to construct multiple candidate examples (e.g., 4 or 6), enabling more diverse comparisons. From these, we select the candidate with the highest reward score as the `win` example and the one with the lowest score as the `lose` example. As shown in Figure 12 (first and second columns), this approach proves effective, as it allows DreamDPO to mine more positive samples and form higher-quality positive-negative pairs. Second, we evaluate the impact of the LMM scale by conducting experiments with three models: Qwen2.5-VL-3B, Qwen2.5-VL-32B, and Qwen-

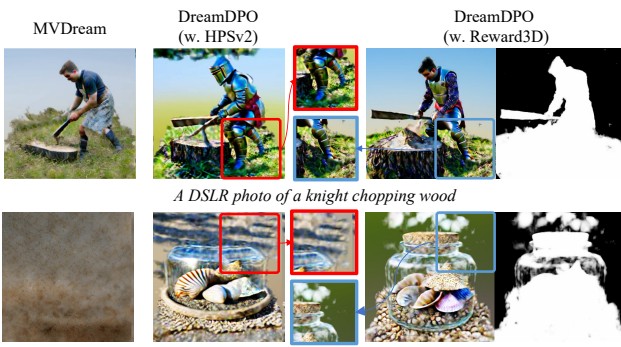

*A DSLR photo of a knight chopping wood*

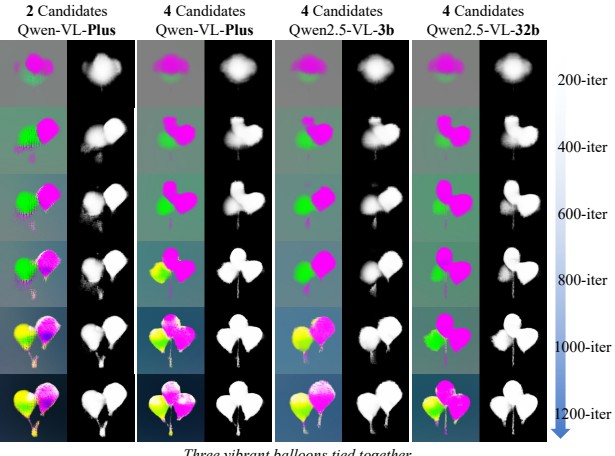

*Various hollow, asymmetrical, textured seashells, collected in a sand-filled, clear glass jar with a twine-tied neck, displayed on a windowsill*

*Figure 11.* The analysis of the 3D reward models. DreamDPO is flexible with both 2D and 3D rewards. The 3D reward model (*e.g.*, Reward3D (Ye et al., 2025)) effectively mitigates Janus issues and some potential artifacts.

*Three vibrant balloons tied together*

*Figure 12.* The analysis of the LMM-based reward model. The results show that increasing the number of comparison candidates helps to better mine positive samples, while high-performing LMMs could enhance DreamDPO's performance.

VL-Plus (Bai et al., 2023). Notably, Qwen2.5-VL-3B, with only 3B parameters, is a relatively weak model. Results in Figure 12 (third column) indicate that when the capability of LMM is limited (*e.g.*, Qwen2.5-VL-3B), it struggles with tasks such as number correction. In SDS-based optimization, early correction of numerical content is crucial, which requires a strong LMM. Thus, larger and more capable LMMs can enhance DreamDPO's performance.

**Further application.** To showcase the potential of our method, we provide empirical results on text-to-avatar generation. Specifically, we replace the score sampling loss in HeadStudio (Zhou et al., 2024) which is a 3DGS (Kerbl et al., 2023) based avatar generation framework, with our reward loss. As illustrated in Figure 13, our method achieves great generation results. This underscores the broader applicability of our method to various generation tasks, *e.g.*, 4D generation (Bahmani et al., 2024) and scene genera-

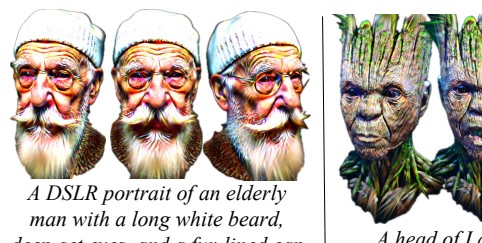

*A DSLR portrait of an elderly man with a long white beard, deep-set eyes, and a fur-lined cap*

*A head of I am Groot*

*Figure 13.* The further application of DreamDPO. We conduct toy experiments on text-to-avatar generation by combining DreamDPO with Gaussian-based avatar generation framework (Zhou et al., 2024). More details can be checked in Appendix B.3.

tion (Zhang et al., 2024a), *etc*.

## 5. Conclusion

In this work, we propose DreamDPO, an optimization-based 3D generation method that offers human preferences and fine-grained control for generation. The method is built on three key steps: pairwise example construction, pairwise example comparison, and preference-guided optimization. Unlike existing methods that rely on precise pointwise quality evaluations, DreamDPO uses pairwise comparison informed by reward models or large multimodal models. It enables a more flexible optimization process. By incorporating human preferences directly into the optimization, DreamDPO generates 3D assets that are better aligned with input text and exhibit enhanced texture and geometry quality. Comprehensive experimental results demonstrate that DreamDPO surpasses previous state-of-the-art methods in both output quality and controllability. Lastly, we hope DreamDPO paves the way for more refined, adaptable, and human-aligned 3D content generation solutions.

**Limitations and future work.** While DreamDPO has shown improvements in aligning 3D generation with human preferences, several avenues for future research could further enhance its performance and applicability. The primary limitations of DreamDPO are as follows: (1) AI feedback can be used for guidelines, but is largely limited to the inherent power of generative models. (2) Open API can provide more freedom, but actually brings more instability, where instruction prompts should be designed carefully.

To address these limitations, we suggest the following directions for future work: (1) Enhancing generative models. Incorporating image prompts (Chen et al., 2024) to introduce explicit guidance could improve alignment with user expectations by providing a more detailed context for generation. (2) Improving the robustness of models in pairwise comparison. Exploring prompt-free methods, such as using object detection models (Wang et al., 2023c) or grounding models (Oquab et al., 2023), for number and attribute correction, might reduce dependencies on prompt design.

## Acknowledgments

This work was supported in part by the National Natural Science Foundation of China (U2336212), Fundamental Research Funds for the Zhejiang Provincial Universities (226-2024-00208), Earth System Big Data Platform of the School of Earth Sciences, Zhejiang University. Xiaobo Xia is supported by MoE Key Laboratory of Brain-inspired Intelligent Perception and Cognition, University of Science and Technology of China (Grant No. 2421002).

This research/project is supported by the National Research Foundation, Singapore under its National Large Language Models Funding Initiative (AISG Award No: AISG-NMLP-2024-002). Any opinions, findings and conclusions or recommendations expressed in this material are those of the author(s) and do not reflect the views of National Research Foundation, Singapore.

## Impact Statement

This paper advances the field of text-to-3D generation by introducing DreamDPO, an optimization-based framework that integrates human preferences into the 3D generation process. By aligning generated content more closely with human preferences, our approach enhances the quality, controllability, and applicability of 3D content creation across various domains, including gaming, virtual reality, design, and digital media.

From an ethical perspective, our work raises considerations related to bias in human preference data, potential misuse in deepfake or unauthorized content generation, and the environmental impact of computationally intensive optimization processes. To mitigate these risks, we promote transparency by open-sourcing our code and models, allowing the research community to further evaluate and refine our approach. Additionally, we encourage responsible use of preference-driven generation techniques to ensure ethical applications.

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

# Appendix

# A. Related Work

## A.1. Text-to-Image Generation

With the development of vision-language models (Radford et al., 2021) and diffusion models (Sohl-Dickstein et al., 2015; Ho et al., 2020), great advancements recently have been made in text-to-image generation (Nichol et al., 2021; Ho et al., 2022; Zhang et al., 2023b). In particular, Stable Diffusion (Rombach et al., 2022b) is a notable framework that trains the diffusion models on latent space, leading to reduced complexity and detailed preservation. In addition, with the emergence of text-to-2D models, more applications have been developed (Liu et al., 2023a; 2025; 2024c;b), *e.g.*, spatial control (Voynov et al., 2023; Zhang et al., 2023c), concept control (Gal et al., 2022; Ruiz et al., 2022), and image editing (Brooks et al., 2023).

## A.2. Text-to-3D Generation

The success of the 2D generation is incredible. However, it is challenging to transfer image diffusion models to 3D because of the difficulty of 3D data collection. Fortunately, Neural Radiance Fields (NeRF) (Mildenhall et al., 2020; Barron et al., 2022) provided new insights for 3D-aware generation, where only 2D multi-view images are needed in 3D scene reconstruction. Combining prior knowledge from text-to-2D models, several methods, such as DreamField (Jain et al., 2022), DreamFusion (Poole et al., 2022), and SJC (Wang et al., 2022), have been proposed to generate 3D objects guided by text prompts (Li et al., 2023b). However, the vanilla score distillation sampling loss (Poole et al., 2022) suffers from issues such as over-saturation, over-smoothing, and Janus problems, *etc*. Recently, several works have proposed improvements to enhance generation quality (Wang et al., 2024; Yu et al., 2023; Zhu et al., 2023; Katzir et al., 2023; Chung et al., 2024; Wu et al., 2024b; Zhuo et al., 2024), and inspire multiple applications that include but are not limited to text-guided scene generation (Cohen-Bar et al., 2023; Höllein et al., 2023), text-guided 3D editing (Haque et al., 2023; Kamata et al., 2023), and text-guided avatar generation (Cao et al., 2023; Jiang et al., 2023; Han et al., 2023; Zhou et al., 2024). Additionally, with the availability of large 3D datasets (Deitke et al., 2023; 2024), some works (Liu et al., 2023b; Shi et al., 2023; Liu et al., 2024a; 2023c; Long et al., 2024) leverage multi-view information to address the Janus problem more effectively.

Recent advancements in multi-view diffusion models have enabled the development of large reconstruction models (Hong et al., 2023; Tang et al., 2024; Zou et al., 2024) that can generate 3D objects from multi-view inputs within minutes. In parallel, 3D generative models (Zhang et al., 2023a; Wang et al., 2023b; Zhang et al., 2024b; Xiang et al., 2025) have emerged, enabling the direct creation of 3D objects guided by text or images. These feed-forward methods significantly accelerate the 3D generation process, fostering broader applications in areas such as gaming and 3D printing. Aligning feed-forward methods with human preferences is meaningful but remains underexplored, primarily due to the significant computational resources required. DreamDPO offers valuable insights for feed-forward methods. Specifically, by constructing candidates with varying noise and using ranking-guided optimization, DreamDPO provides a potential pathway for reinforcement learning-based optimization of feed-forward methods.

## A.3. Learning from Human Preferences

Learning from human preferences is essential for improving the alignment and performance of generative models across various domains, including large language models (LLMs) (Bai et al., 2022a;b; Lee et al., 2023a; Zhao et al., 2023; Su et al., 2025), text-to-image diffusion models (Black et al., 2023; Lee et al., 2023b; Clark et al., 2023; Xu et al., 2024b; Wallace et al., 2024; Fan et al., 2024; Zhang et al., 2024c), and text-to-3D generation (Xie et al., 2024; Ye et al., 2025). Reinforcement Learning from Human Feedback (RLHF) (Christiano et al., 2017) has been proven effective in refining LLMs to better align with human preferences. It often involves collecting human feedback datasets, training a reward model, and fine-tuning the language model using reinforcement learning. InstructGPT (Ouyang et al., 2022) is a notable work that employs a two-stage fine-tuning strategy to align GPT-3 with human instructions, which leads to more coherent and contextually appropriate outputs.

The success of RLHF in LLMs has inspired the applications in text-to-image diffusion models to enhance image generation quality and align with human preferences. In more detail, RWR (Lee et al., 2023b) first introduces human feedback-based reward fine-tuning for diffusion models, which fine-tunes Stable Diffusion (Rombach et al., 2022a) using log probabilities of the denoising process. ImageReward (Xu et al., 2024b) proposes a reward model specifically for text-to-image tasks and further develops reward feedback Learning for refining diffusion models. DiffusionDPO (Wallace et al., 2024) uses DPO to optimize diffusion models using human comparative data, while DPOK (Fan et al., 2024) integrates policy optimization

with KL regularization for improved alignment. Recently, advancements in multi-view diffusion models have facilitated significant progress in text-to-3D generation. For instance, Carve3D (Xie et al., 2024) enhances text-to-3D generation with a Multi-view Reconstruction Consistency (MRC) metric for improved consistency and quality. DreamReward (Ye et al., 2025) improves text-to-3D models using human feedback. It collects a 3D dataset with human annotations, trains Reward3D as a multi-view reward model, and introduces DreamFL to create 3D assets aligned with human preferences. However, these works rely heavily on large-scale datasets to train a reward model or utilize it as preference feedback, which is very expensive for 3D generation.

# B. Additional Implementation Details

## B.1. Pseudo-Code for DreamDPO

A more detailed pseudo-code for DreamDPO is presented in Algorithm 1.

---

**Algorithm 1** Pseudo-code for DreamDPO

---

**Input:** Text-to-image diffusion model $\epsilon_\phi$. Ranking model $r$. Learning rate $\eta$ for 3D representation parameters. A prompt $y$. Evaluating threshold of score gap $\tau$.

1: Initialization A 3D representation presenting with NeRF $\theta$
2: **while** not converged **do**
3:     Randomly sample a camera pose $c$, pairwise 2D noise $\epsilon^1$ and $\epsilon^2$, and timestep $t \sim \text{Uniform}(\{0, \cdots, T\})$.
4:     Render at pose $c$ to get a image $\mathbf{x}_0$.
5:     Add noise $\epsilon^1$ and $\epsilon^2$ to $\mathbf{x}_0$ and get $\mathbf{x}_t^1$ and $\mathbf{x}_t^2$, respectively.
6:     Denoise with the predicting noise:
$$\hat{\mathbf{x}}_0^1 = \frac{\mathbf{x}_t^1 - \sigma_t \epsilon_\theta(\mathbf{x}_t^1; y, t)}{\alpha_t},$$
$$\hat{\mathbf{x}}_0^2 = \frac{\mathbf{x}_t^2 - \sigma_t \epsilon_\theta(\mathbf{x}_t^2; y, t)}{\alpha_t}.$$
7:     Score the prediction $(\hat{\mathbf{x}}_0^1, \hat{\mathbf{x}}_0^2)$ online via a rank model $r$, yielding the pairwise comparison $(\mathbf{x}_t^{\text{win}}, \mathbf{x}_t^{\text{lose}})$.
8:     Compute the score gap:
$$s_{\text{gap}} = r(\mathbf{x}_t^{\text{win}}, y) - r(\mathbf{x}_t^{\text{lose}}, y). \tag{7}$$
9:     **if** $s_{\text{gap}} < \tau$ **then**
10:
$$\nabla_\theta \mathcal{L}_{\text{Reward}} = \mathbb{E}_t \left[ w(t) \left( \epsilon_\phi^s(\mathbf{x}_t^{\text{win}}; y, t) \right) \frac{\partial \mathbf{x}}{\partial \theta} \right],$$
11:     **else**
12:
$$\nabla_\theta \mathcal{L}_{\text{Reward}} = \mathbb{E}_t \left[ w(t) \left( \left( \epsilon_\phi^s(\mathbf{x}_t^{\text{win}}; y, t) - \epsilon^{\text{win}} \right) - \left( \epsilon_\phi^1(\mathbf{x}_t^{\text{lose}}; y, t) - \epsilon^{\text{lose}} \right) \right) \frac{\partial \mathbf{x}}{\partial \theta} \right].$$
13:     **end if**
13:     $\theta \leftarrow \theta - \eta \nabla_\theta \mathcal{L}_{\text{Reward}}$.
14: **end while**

---

## B.2. Details of LMM-based Pairwise Comparison

We detail the implementation of the LMM-based pairwise comparison. We use the large visual-language model "qwen-vl-plus-latest" from QwenVL (Bai et al., 2023) as the default LMM. Given pairwise examples, we conduct the comparison query sequentially. For each query, we first insert a predefined "yes" or "no" question into the comparison prompt, such as "*Is the leaf shouting?*" for the prompt "*A shouting leaf.*" Then, the LMM performs visual question answering based on the provided image and query. Finally, we extract the number of "yes" responses as the score. The following prompts are used for the queries:

---

**Comparison Query**

[Task Description]: You are an expert in evaluating the alignment between a given text description and an image. Your task is to answer each of the alignment questions with either "Yes" or "No" based on the image. Provide your responses in the format specified below.

[Evaluation Instruction]:

1. Carefully analyze the provided image and answer questions based on the image.

2. For each question, answer with either "Yes" or "No". Do not provide explanations or additional information.

[Evaluation Question(s)]:

Q1: {Question}

...

[Output Format]:

A1: [Yes/No]

...

---

### B.3. Details of Text-to-Avatar Generation

We detail the toy exploration of text-to-avatar generation using DreamDPO. Specifically, we integrate the reward loss into HeadStudio (Zhou et al., 2024), a Gaussian-based avatar generation framework. HeadStudio is an optimization-based method that utilizes a score sampling loss (Katzir et al., 2023) with ControlNet (Zhang et al., 2023c) to optimize an animatable head prior model. By replacing the score sampling loss with the reward loss, we leverage ControlNet to generate avatars.

## C. Supplementary Experimental Settings

### C.1. Details of Measurement Metrics

In the main paper, we employ two evaluation strategies to demonstrate the superiority of the proposed method. Here we supplementary the details of the measurements.

**Evaluation with ImageReward.** ImageReward (Xu et al., 2024b) is a text-to-image human preference reward model. Due to its effectiveness, it has been broadly used for human preference evaluation in text-to-image generation (Fan et al., 2024) and text-to-3D generation (Ye et al., 2025). Given a (text, image) pair, it extracts image and text features, combines them with cross-attention, and uses an MLP to generate a scalar for preference comparison. For each 3D asset, we uniformly render 120 RGB images from different viewpoints. Afterward, the ImageReward score is computed from the multi-view renderings and averaged for each prompt.

**Evaluation with GPTEval3D.** We utilize GPTEval3D (Wu et al., 2024a), which is a comprehensive benchmark for text-to-3D generation evaluation. GPTEval3D includes 13 baseline methods $\mathcal{M}$, 110 text prompts, and 5 criteria that are text-asset alignment, 3D plausibility, texture details, geometry details, and texture-geometry coherency respectively. For a new method, GPTEval3D employs GPT-4V to compare 3D assets generated by this new method and one of the baseline methods with the same input text prompt. These pairwise comparison results are then used to calculate the Elo rating for each model. Specifically, let $\mathbf{A}$ be a matrix where $\mathbf{A}_{ij}$ represents the number of times that the $i$-th model outperforms the $j$-th model in comparisons. The Elo ratings for the models are computed by optimizing the following objective:

$$\sigma = \arg\max_{\sigma} \sum_{i \neq j} \mathbf{A}_{ij} \log \left( 1 + 10^{(\sigma_j - \sigma_i)/400} \right),\tag{8}$$

where $\sigma_i \in \mathbb{R}$ is the Elo rating of the $i$-th model. In this work, we calculate Elo ratings within the existing tournament, initializing, and freezing baseline scores as specified in the official code[2]. For interested readers, please refer to (Wu et al., 2024a).

MVDream       DreamDPO (10000 Steps)       DreamDPO (4000 steps)

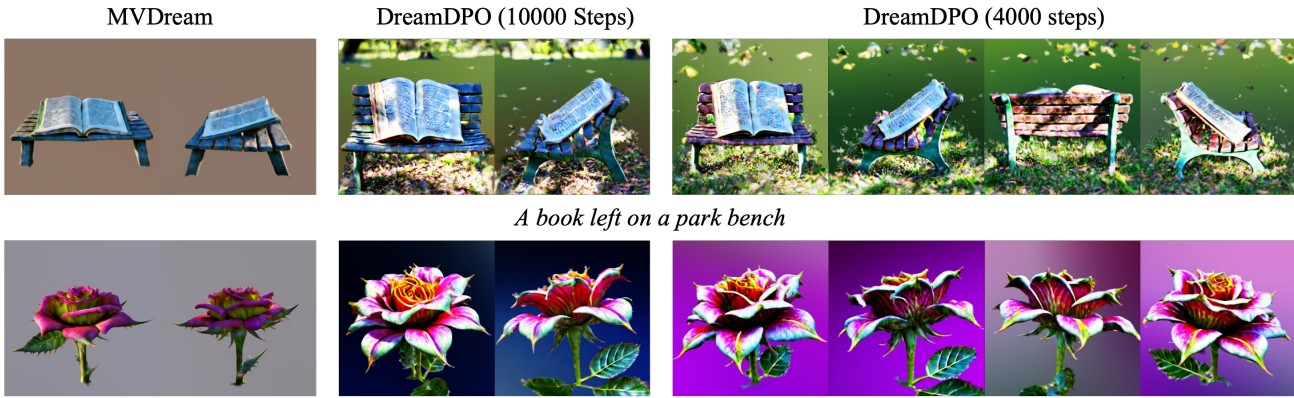

*A book left on a park bench*

*A thorny rose*

*Figure 14.* The analysis of reducing generation time. DreamDPO could reduce generation time by 25% (to 1.5 hours) while maintaining improved performance.

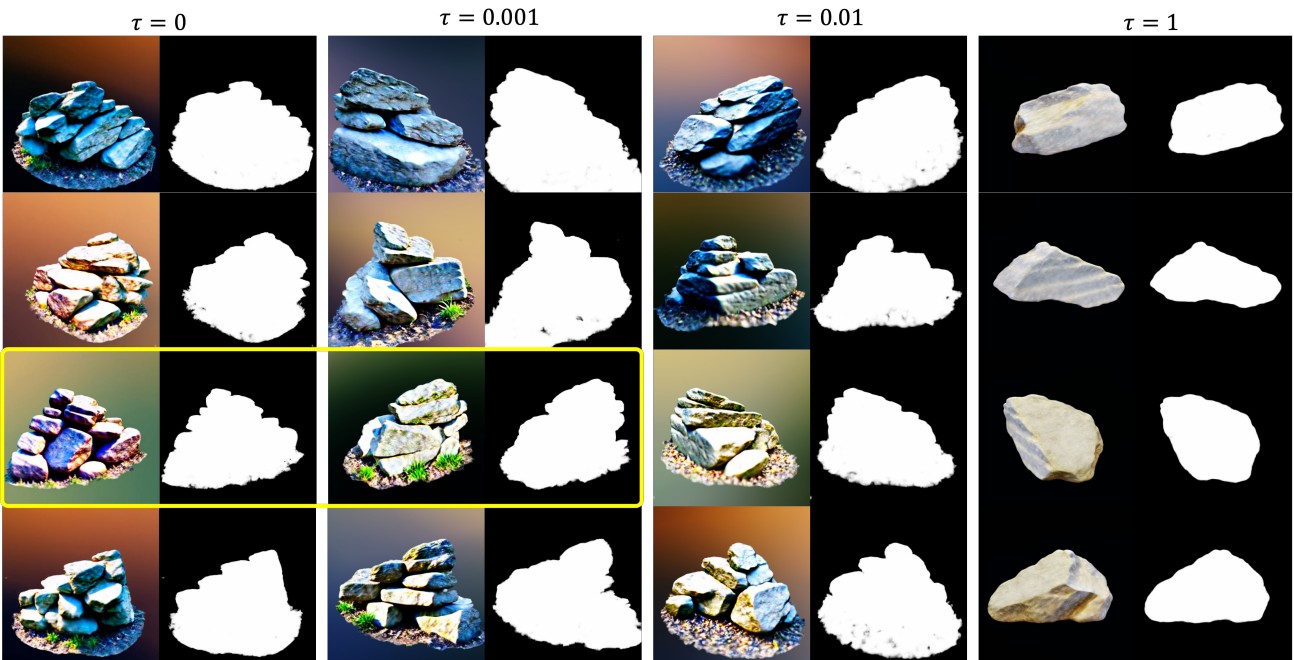

*Figure 15.* The analysis of the score gap threshold $\tau$. We conduct 3D experiments with $\tau$ ranging from 1 to 0. The results indicate that $\tau = 0$ results in over-saturation, and a small but non-zero $\tau$ leads to more detailed outputs.

## D. Supplementary Experimental Results

### D.1. Discussion

**Discussion on training time.** The vanilla SDS takes around 1 hour to optimize. Our method takes approximately 2 hours due to the pairwise example construction. To speed up the process, we can adopt a simple yet effective strategy: performing SDS for the first 6000 iterations and then switching to DreamDPO. As shown in Figure 14, this approach reduces the generation time by 25% (to around 1.5 hours) while maintaining improved performance.

**Discussion on the score gap threshold in the 3D setting.** We present the ablation study of the score gap threshold $\tau$ in the 3D setting (see Figure 15). The results indicate that a large $\tau$ (*e.g.*, $\tau = 1$) degrades DreamDPO to the SDS loss, resulting in an over-smoothing issue. Conversely, setting $\tau = 0$ may result in over-saturation in some cases, such as a purplish sunlight appearance in rocks. Therefore, we recommend using a small but non-zero $\tau$.

---

[2]https://github.com/3DTopia/GPTEval3D/blob/main/data/tournament-v0/config.json

## D.2. More Qualitative Results

We present additional qualitative results in Figure 16 and Figure 17. The comparisons demonstrate that our method generates human-preferred 3D assets, with improved text alignment and enhanced geometric and texture details.

MVDream                                    **DreamDPO (Ours)**

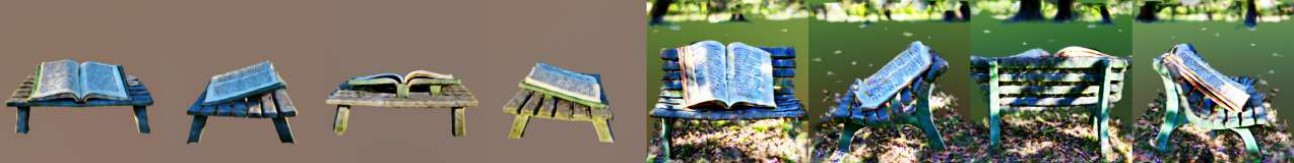

*A book left on a park bench*

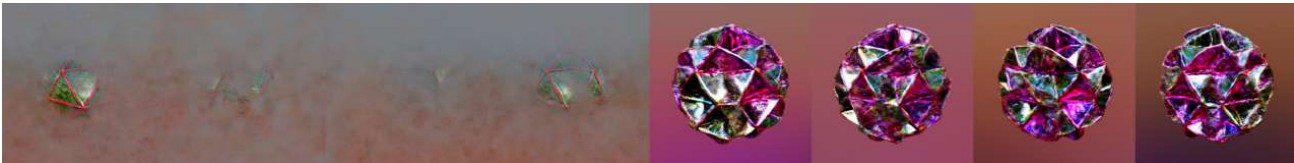

*A small, solid, geometrically spherical, metallic orb, with a glossy ruby finish, nestled in a nest of black velvet, untouched and gleaming under a spotlight*

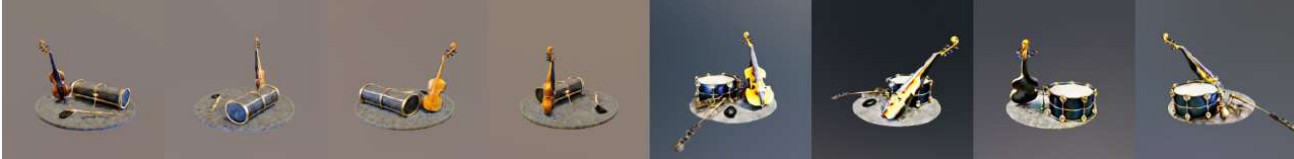

*An ensemble of hollow, irregularly shaped musical instruments, including a saxophone, a violin, and a drum, resting on a stage before a jazz concert"*

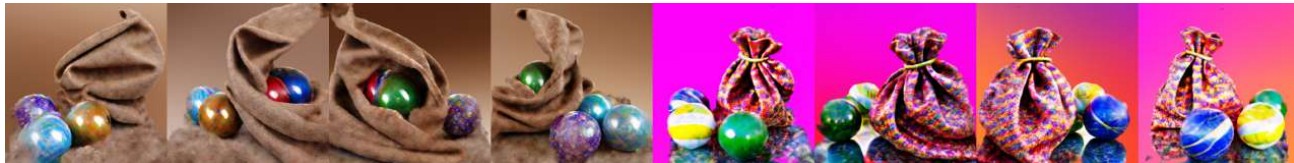

*A bunch of colorful marbles spilling out of a red velvet bag*

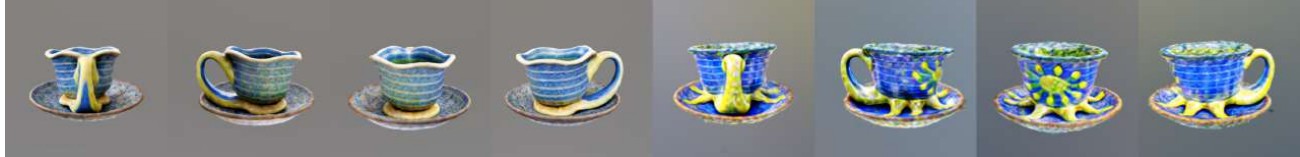

*A ceramic upside down yellow octopus holding a blue green ceramic cup*

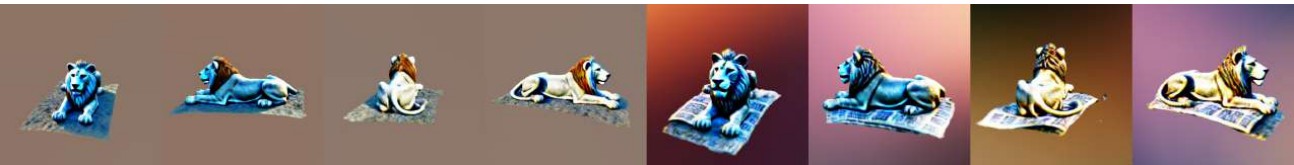

*A DSLR photo of a lion reading the newspaper*

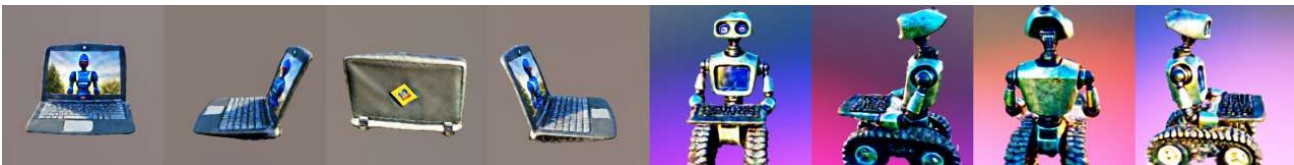

*A DSLR photo of a humanoid robot using a laptop*

*Figure 16.* More qualitative results using DreamDPO.

*MVDream*                    **DreamDPO (Ours)**

*A DSLR photo of a blue jay standing on a large basket of rainbow macarons*

*A DSLR photo of a cat lying on its side batting at a ball of yarn*

*A DSLR photo of a cat magician making a white dove appear*

*A DSLR photo of a knight chopping wood*

*A beagle in a detective's outfit*

*A collection of solid, irregularly shaped hand tools, with wooden handles and metal ends, well-used and slightly rusty, hanging on a pegboard in a workshop*

*A DSLR photo of a basil plant*

*Figure 17.* More qualitative results using DreamDPO.

