# OpenReview forum: "DreamDPO: Aligning Text-to-3D Generation with Human Preferences via Direct Preference Optimization"
_ICML.cc/2025/Conference — ICML 2025 poster_

### Official Review · Reviewer_kB21 · 2025-03-08

**Overall Recommendation:** 3

**Summary:**

This paper introduces DreamDPO, an optimization-based framework for text-to-3D generation that aligns the generated 3D content with human preferences. Instead of relying on absolute quality scores from reward models, DreamDPO employs Direct Preference Optimization (DPO). The method operates in three iterative steps: 1) Pairwise Example Construction, 2) Pairwise Comparison, and 3) Preference-Guided Optimization, where a novel piecewise loss function, derived from the pairwise preference, guides the update of the 3D representation parameters. This piecewise loss is designed to prevent noisy gradients when the compared examples are very similar.

The paper demonstrates the proposed method's effectiveness through experiments on the GPTEval3D benchmark, comparing it against 13 existing text-to-3D generation methods. Quantitative evaluations using ImageReward, CLIP score, and GPTEval3D's metrics to show good results in terms of text-image alignment, 3D plausibility, and texture/geometry detail. Qualitative comparisons also support these claims. The paper also explores the use of MLLMs/LMMs for providing preference feedback. Ablation studies investigate the impact of different diffusion model backbones, reward models, and the score gap threshold used in the piecewise loss.


## Update After Rebuttal
I thank the authors for their detailed rebuttal. The provided quantitative comparison with DreamReward (Rebuttal Table 2-1) is appreciated and clarifies their relative performance. The authors also committed to better contextualizing their work regarding feed-forward methods.

However, the justification for the $\tau$ hyperparameter remains unclear, and the potential limitations of using 2D metrics for 3D evaluation were not fully resolved. Most importantly, the concern regarding the method's slow computational speed compared to state-of-the-art feed-forward approaches was not addressed.

Due to these remaining issues, particularly the unaddressed speed limitation, my assessment has not changed.

**Claims And Evidence:**

Overall, most of the claims made in the paper are backed up by good evidence. The results on the GPTEval3D benchmark (Table 1) and the visual examples (Figures 2, 3, 7, 11, 12) show that DreamDPO can generate reasonable 3D objects that match the text descriptions well. The piecewise loss function also seems to be important, as shown by Figure 6 and the explanation in Section 3.1. The experiments in Section 4.3 and Figure 8 demonstrate the flexibility of the method.

The paper does compare DreamDPO to DreamReward, but only qualitatively (Figure 7). While these visual comparisons are helpful, it would be much stronger to also include a quantitative comparison. This is a missed opportunity to directly compare DreamDPO to a closely related method that uses a different approach (a custom-trained 3D reward model).

**Essential References Not Discussed:**

I did not find any essential references that were missing from the paper's discussion.

**Experimental Designs Or Analyses:**

The use of the GPTEval3D benchmark and multiple metrics is generally a good approach, and the comparisons to a wide range of baselines are valuable. Still, the lack of a quantitative comparison with DreamReward is a significant issue. The ablation studies are helpful in understanding the impact of different components, but the exploration of LMM capabilities could be more extensive, and the justification of the $tau$ relies more on the 2D toy example. While the qualitative evaluations provide visual support, they are inherently subjective. Maybe a more thorough user study could strengthen the claims of improved human preference alignment.

**Methods And Evaluation Criteria:**

The methods proposed in this paper show some promise for text-to-3D generation. The DreamDPO framework and the piecewise loss function are interesting ideas. However, there are some aspects of both the methods and the evaluation that could be strengthened. The reliance on pairwise comparisons and 2D image metrics, while common practice, introduces potential limitations that aren't fully explored, such as the gap between 2D-3D perception. More thorough ablation studies focusing on the piecewise loss would be beneficial. The evaluation, while using a standard benchmark, could be significantly improved by including direct 3D mesh quality metrics and, crucially, by providing a quantitative comparison with DreamReward. The current evaluation doesn't provide fully convincing evidence of the method's superiority, especially considering the reliance on rendered 2D views only.

**Other Comments Or Suggestions:**

It would be helpful for the paper to better contextualize its approach within the broader landscape of 3D generation. While the SDS-based method using 2D diffusion models is a valid area of research, many of the current state-of-the-art results in terms of visual quality and speed are coming from feed-forward methods that use large-scale 3D models (diffusion or autoregressive) trained on massive 3D datasets. Mentioning these other approaches, and explaining why this paper focuses on the SDS-based method, would make the paper more complete.

**Other Strengths And Weaknesses:**

In addition to the points already raised, there are a few minor weaknesses. The discussion of limitations could be more thorough, particularly regarding the reliance on pre-trained models and 2D metrics. While the experiments analyze the impact, more justification about the hyperparameter selection would also make the paper clearer and stronger.

Crucially, the paper is very slow compared to many state-of-the-art 3D generation methods that use feed-forward approaches. This is a major practical limitation that needs to be addressed more directly.

**Questions For Authors:**

1. The paper is using 2D image reward models (HPSv2, ImageReward) to evaluate 3D assets. Could you elaborate on the potential limitations of this approach? Are there specific types of 3D inconsistencies or artifacts that these 2D models might miss? Have you considered any ways to mitigate this domain gap (e.g., incorporating some form of 3D consistency check)?

2. The piecewise loss uses a threshold of $tau = 0.001$. Figure 6 shows this works in 2D, but could you give a bit more explanation, or maybe some 3D-specific results, to justify this particular value?

3. The paper employs MVDream as the backbone. Have you tried other SDS-based text-to-3D generation methods? How does the proposed method perform with them?

**Relation To Broader Scientific Literature:**

This paper connects its contributions to the broader literature on (SDS-based) text-to-3D generation and learning from human preferences.

**Theoretical Claims:**

This paper is primarily empirical and does not present theoretical claims or proofs.

---

> ### Author Rebuttal · Authors · 2025-04-01
>
> We thank the reviewer for the positive reviews. We provide our responses below.
>
> > **Q1:** Quantitative comparisons with DreamReward.
>
> **A1:** Thanks for your suggestion. We conduct a quantitative comparison on GPTEval3D to evaluate human preference, including the correction of numbers. In specific. we report the number accuracy and CLIPScore. The results below show that DreamDPO demonstrates competitive performance in prompt alignment and significant improvement in number correction.
>
> | Method       | Number Accuracy $\uparrow$ | CLIPScore $\uparrow$ |
> | ----------- | --------------- | ------------------ |
> | DreamReward | 41.7%             | 0.2855           |
> | DreamDPO    | 71.7%             | 0.2787           |
>
> Table 2-1: Quantitative comparisons of DreamDPO and DreamReward on GPTEval3D. We calculate the CLIPScore for prompt alignment and number accuracy for number correction.
>
> We attribute this to that DreamDPO does not rely on precise reward scores, allowing it to leverage various black-box AI models for scoring, which helps correct numbers and attributes and improves human preference alignment.
>
> > **Q2:** The potential limitations on 2D metrics.
>
> **A2:** Yes. 2D image reward models are relatively mature, with abundant data and good reward performance. However, they have limitations. While they improve prompt alignment, such as correcting character attributes (e.g., changing "man" to "knight"), they struggle to address inherent Janus issues (e.g., characters with three legs). Our DreamDPO is flexible with both 2D and 3D rewards. Experiments in [Figure 3-1](https://imgur.com/a/XdVx2sn) demonstrate that it effectively mitigates Janus issues and some potential artifacts, offering a more reliable evaluation for 3D consistency.
>
> > **Q3:** More exploration of LMM capabilities.
>
> **A3:** Thanks. We have further analyzed the failure cases of DreamDPO with LMMs (see R2-Q2). Additionally, we demonstrate that increasing the number of comparison candidates can effectively mine positive samples, leading to improved performance with LMMs (see R2-Q3). Lastly, we show that DreamDPO works effectively across various LMMs (see R2-Q4).
>
> > **Q4:** Additional discussion with feed-forward methods.
>
> **A4:** Thanks. Aligning feed-forward methods with human preferences is meaningful but remains underexplored, primarily due to the significant computational resources required. DreamDPO offers valuable insights for feed-forward methods. Specifically, by constructing candidates with varying noise and using ranking-guided optimization, DreamDPO provides a potential pathway for RL optimization of feed-forward methods. We will polish the paper with better contextualize within the broader landscape of 3D generation.
>
> > **Q5:** Ablation studies on the score gap threshold in the 3D setting.
>
> **A5:** Thanks. Please kindly check R1-Q1.
>
> > **Q6:** More results with other SDS-based text-to-3D generation methods.
>
> **A6:** Thanks. We evaluate the performance of DreamDPO with different backbones in Section 4.3. In specific, we evaluate the performance of DreamDPO with Stable Diffusion v2.1 (SD2.1). The results demonstrate that DreamDPO works effectively with SD2.1 and achieves competitive results compared to ProlificDreamer. While SD2.1 shows improvements over the baseline, MVDream outperforms due to its superior 3D consistency. Therefore, we adopt MVDream as the default backbone.

---

> > ### Comment · Reviewer_kB21 · 2025-04-05
> >
> > Thank you to the authors for their rebuttal. I have carefully reviewed the rebuttal along with the comments from other reviewers. While I appreciate the clarifications provided, I will be maintaining my original score.

---

### Official Review · Reviewer_yvaT · 2025-03-09

**Overall Recommendation:** 4

**Summary:**

This paper proposes DreamDPO, an optimization-based method to better align 3D generation with human preferences for text-to-3D generation. In detail, it constructs pairwise examples to formulate a reward loss function for preferred images with lower loss and less preferred images with higher loss. It conducts comprehensive experiments with 13 baselines and shows the superiority of the proposed method.

**Claims And Evidence:**

Yes, the claims are clear and convincing.

**Essential References Not Discussed:**

N/A

**Experimental Designs Or Analyses:**

I checked the soundness of the experimental designs and analyses. The main experiment demonstrates the effeciency of the proposed method compared with the other baselines with more delicate 3D generation. The authors also discussed the effect of different backbones, reward models, score gaps, pair examples, model design, and further applications in the experiment parts. I think the experiment is comprehensive and meaningful.

**Methods And Evaluation Criteria:**

Yes, the author utilizes ImageReward score for human preference evaluation, CLIP score for text-image alignment evaluation, and GPTEval3D for 3D quality evaluation to evaluate their proposed method. Their method is effecient by using those meaningful evaluation criteria.

**Other Comments Or Suggestions:**

Please see the weaknesses.

**Other Strengths And Weaknesses:**

* Strengths:
1. This paper is well-written and well-organized.
2. The research topic is insteresting and meaningful. Text-to-3D can be utilized in many practical applications such as games and designing.
3. The proposed method, DreamDPO, is much more efficient than existing baselines, which generates more delicate 3D images.

* Weaknesses:
1. How about the computational efficiency of the proposed method compared with the other baselines?
2. I am still confused about the determination of $x_t^{win}$ and $x_t^{lose}$. Do you assign the sample with lower loss to $x_t^{win}$ and the sample with higher loss to $x_t^{lose}$?
3. If current 3D iterms generated from text-to-3D generation can be used in manufacturing? I ask this question because I am very curious about the practical application of this field.

**Questions For Authors:**

Please see the weaknesses.

**Relation To Broader Scientific Literature:**

N/A.

**Theoretical Claims:**

This paper does not include the theoretical discussion.

---

> ### Author Rebuttal · Authors · 2025-04-01
>
> We thank the reviewer for the positive reviews. We provide our responses below.
>
> > **Q1:** The computational efficiency comparison to other baselines.
>
> **A1:** Thank you for your question. We summarize the computational cost of our method compared with other text-to-3D generation baselines as follows:
>
> | Method             | DreamFusion | Fantasia3D | Latent-NeRF | ProlificDreamer | MVDream | DreamDPO |
> | ----------------- | ---------- | ---------- | ---------- | --------------- | ------- | -------- |
> | Computation Time   | 1.5 hours  | 1.5 hours  | 30 minutes | 10 hours         | 1 hour  | 2 hours  |
>
> Table 2-1: The analysis of the time consuming of text-to-3D generation methods.
>
> While introducing additional computational overhead due to pairwise example construction, our method achieves the best results on the GPTEval3D Benchmark, demonstrating superior alignment with human preferences. Meanwhile, we can mreduce generation time by 25% (to ~1.5 hours) while maintaining performance (see R1-Q4).
>
> > **Q2:** How are $x_t^{win}$ and $x_t^{lose}$ determined?
>
> **A2:** We utilize a reward model to compute the scores of pairwise examples. The sample with a higher score is regarded as the "win" example ($x_t^{win}$), while the sample with a lower score is regarded as the "lose" example ($x_t^{lose}$).
>
> > **Q3:** Can current text-to-3D generated items be used in real-world manufacturing?
>
> **A3:** Yes. Text-to-3D generation is increasingly suitable for real-world manufacturing, particularly in customized product design. It allows engineers to quickly create 3D models from textual descriptions. While further refinement may be needed to meet precise industrial standards, the technology is proving valuable in automating design workflows and enabling mass customization.

---

> > ### Comment · Reviewer_yvaT · 2025-04-03
> >
> > Thank you for the response. I decide to keep my positive score on this paper.

---

### Official Review · Reviewer_YAnn · 2025-03-11

**Overall Recommendation:** 4

**Summary:**

This paper introduces DreamDPO, an optimization-based framework for text-to-3D generation. The authors propose to integrate human preferences into the generation process through direct preference optimization. The authors claim that the key innovation is leveraging pairwise comparisons to guide optimization instead of relying on absolute, pointwise quality scores. In specific, DreamDPO constructs pairwise examples, evaluates them using reward models or large multimodal models (LMM), and then optimizes the 3D content with a designed preference-driven loss function. This paper shows superior alignment with textual inputs and improved controllability over existing methods on established benchmark.

**Claims And Evidence:**

The claims presented are generally supported by substantial evidence through qualitative and quantitative experiments. The experiments demonstrate that DreamDPO somewhat generates higher-quality and more controllable 3D assets compared to existing methods, supported by human preference evaluation scores, GPTEval3D scores, and comprehensive ablation studies.

**Essential References Not Discussed:**

The paper appears comprehensive in its references.

**Experimental Designs Or Analyses:**

The experimental designs and analyses appear sound and rigorous. Detailed ablation studies (e.g., different Gaussian noise, score gap thresholds, reward models, and backbones) are clearly conducted, strengthening the validity of the results. The experimental framework effectively validates the method’s key contributions.

**Methods And Evaluation Criteria:**

The proposed methods and evaluation criteria are sensible and appropriate. The benchmark GPTEval3D dataset is commonly used in text-to-3D generation. Meanwhile, the human preference evaluation using ImageReward also provides robust evaluation metrics. The experimental setups align with standard practices for evaluating generative models, ensuring validity and comparability with prior works.

**Other Comments Or Suggestions:**

The authors do not report the CLIP score in Table 1, which should be removed accordingly in the caption.

**Other Strengths And Weaknesses:**

Pros:
1. The motivation is clear, which addresses the need for improved human alignment and controllability in 3D generation tasks. Besides, the writing is easy to follow.
2. The method is simple but effective. It integrates human preferences through pairwise comparisons, which improves the alignment of the generated 3D content with user expectations and instructions. l believe it can improve other 3D generation methods based on optimization and have a wide generalizability.
3. The experiments are convincing. Experiments show DreamDPO generates higher-quality outputs in terms of geometric and textural details compared to baseline models like DreamFusion, DreamGaussian, and MVDream. Besides, it offers explicit and fine-grained control over attributes.

Cons:
1.	The ablation study of the score gap threshold $\tau$ in the 3D generation setting is lacking, limiting insights into its influence on performance.
2.	The analysis of failure cases is insufficient, which constrains a deeper understanding of DreamDPO’s limitations and operational boundaries.
3.	The potential for performance improvement through increasing the number of comparison candidates is worth exploring and should be considered in future work.
4.	The applications of various large multimodal models are underexplored. A more detailed discussion is encouraged to assess how LMM quality affects alignment with human preferences.

**Questions For Authors:**

1.	The current parameter analysis is conducted on 2D (Figure 6). To provide a more comprehensive evolution, I suggest including ablation studies on the score gap threshold $\tau$ in the 3D setting.
2.	Including failure cases would offer valuable insights into the limitations of DreamDPO and help delineate the boundaries of its effectiveness.
3.	While the pairwise comparison method is effective, its potential could be further explored by increasing the number of comparison candidates per iteration. For example, investigating group-based loss functions, such as GRPO loss in DeepSeek-R1, might lead to improved performance.
4.	Although the authors acknowledge that fine-grained controllability is currently limited by the capabilities of the LMM, it would be worthwhile to explore whether advancements in LMMs could mitigate this constraint.
5.	It is recommended to conduct a broader evaluation across more LMMs, ranging from weaker to more advanced models, to gain a deeper understanding of their impact on controllability and overall performance.

**Relation To Broader Scientific Literature:**

The most related paper is DreamReward. Both DreamReward and DreamDPO focus on the human preference alignment problem in text-to-3D generation. DreamReward collected and labeled a multi-view preference dataset and accordingly trains a multi-view reward model, and use this model for 3D human preference alignment.
Different with DreamReward, the authors claim that DreamDPO shifts from absolute scoring to direct preference optimization, and introduce a pair-wise optimization loss, which reduce reliance on precise pointwise quality evaluations. Therefore, DreamDPO could use the image-aware reward model and LMM for preference alignment generation.

**Theoretical Claims:**

No explicit theoretical proofs are provided for this paper.

---

> ### Author Rebuttal · Authors · 2025-04-01
>
> We thank the reviewer for the positive reviews. We provide our responses below.
>
> > **Q1:** Ablation studies on the score gap threshold in the 3D setting.
>
> **A1:** We present the ablation study of the score gap threshold $\tau$ in the 3D setting (see [Figure 2-1](https://imgur.com/a/XdVx2sn)). The results indicate that a large $\tau$ (e.g., $\tau=1$) degrades DreamDPO to the SDS loss, resulting in an over-smoothing issue. Conversely, setting $\tau=0$ results in over-saturation, such as a purplish sunlight appearance in rocks. Therefore, we recommend using a small but non-zero $\tau$.
>
>
> > **Q2:** Show some failure cases to highlight DreamDPO’s limitations.
>
> **A2:** While DreamDPO has shown improvements in aligning 3D generation with human preferences, there are still some failure cases. For example, number and attribute correction are crucial for prompt alignment, but DreamDPO sometimes fails to address these issues shown in [Figure 2-2](https://imgur.com/a/XdVx2sn). We found that this limitation arises from the generative model's capacity. When the positive sample is hard to generate, DreamDPO struggles to construct effective positive-negative pairs, causing it to degrade into SDS.
>
> > **Q3:** Increasing the number of comparison candidates per iteration.
>
> **A3:** Thank you for your valuable suggestion. We further extend DreamDPO to multi-sample comparisons. In specific, we expand STEP1 to multi-example construction, enabling the creation of more comparison candidates (e.g., 4 or 6). We then select the candidate with the highest score as the "win" example and the lowest score sample as the "lose" example. Experiments in [Figure 2-3](https://imgur.com/a/XdVx2sn) (the first and second columns) demonstrate that this improvement is effective, as it better mines positive samples and enables DreamDPO to construct positive-negative pairs more effectively.
>
> > **Q4:** More evaluation on DreamDPO with large multi-modal models.
>
> **A4:** Following your recommendation, we evaluated the robustness of DreamDPO with various LMMs. Specifically, we conducted experiments using three large multimodal models: Qwen2.5-VL-3B, Qwen2.5-VL-32B, and Qwen-VL-Plus. Notably, Qwen2.5-VL-3B has only 3B parameters, making it a relatively weak LMM. The results in [Figure 2-3](https://imgur.com/a/XdVx2sn) show that when the LMM performance is relatively poor (e.g., Qwen2.5-VL-3B in third column), it fails to correct numbers effectively. In SDS-based optimization, number correction should perform early, which requires strong LMM capabilities. Therefore, high-performing LMMs can improve DreamDPO.

---

> > ### Comment · Reviewer_YAnn · 2025-04-02
> >
> > Thanks for the authors ‘ reply, I think the author has solved my concerns, and I am willing to raise my score.

---

### Official Review · Reviewer_x6sm · 2025-03-11

**Overall Recommendation:** 3

**Summary:**

The paper introduces the DreamDPO framework, designed to enhance text-to-3D generation by aligning generated content more closely with human preferences. Traditional methods often fall short due to their heavy reliance on precise evaluations, restricting flexibility and applicability. DreamDPO employs an optimization-based approach that incorporates human preferences through direct preference optimization.

The methodology consists of three key steps:

1. **Pairwise Example Construction**: The framework constructs pairwise examples by applying varying Gaussian noise to the diffusion model.
2. **Pairwise comparison**: It uses either reward models or large multimodal models to rank these examples based on their alignment with the provided textual prompts.
3. **Preference-guided optimization**: A preference-driven loss function guides the optimization of the 3D representation.

These steps allow DreamDPO to minimize dependence on exact scoring while granting robust control over the generation process.

**Claims And Evidence:**

The claims made in the paper regarding the DreamDPO framework for text-to-3D generation are well-supported by clear and convincing evidence based on the findings presented.

1. **Performance Claims**: The paper asserts that DreamDPO outperforms existing techniques. This claim is supported by extensive experiments outlined in section 4, which detail experiment setups and comprehensive results comparing DreamDPO with 13 state-of-the-art methods.
2. **Quality and Control Claims**: The paper emphasizes improvements in the texture and geometry quality of the generated 3D assets. Direct reference to qualitative results is made, indicating that the method delivers high-quality outcomes and offers fine-grained control over the generation process. These aspects are discussed through quantitative and qualitative analyses, supplemented by ablation studies that validate the claims regarding quality and adaptability.
3. **Innovative Contributions**: The paper introduces a novel optimization-based approach integrating human preferences via direct preference optimization. This conceptual contribution is bolstered by detailing the three-step optimization process, highlighting its significance and effectiveness in achieving better alignment with human preferences compared to traditional methods.

However, while the paper’s claims are largely supported, additional context or clarification could enhance the arguments surrounding the advantages of using large multimodal models in the optimization loop and the implications of reduced reliance on precise scoring. Providing a more detailed discussion on how these elements distinctly outperform existing approaches could help mitigate doubts regarding the robustness of these claims since large multimodal models sometimes could have poor performance.

Overall, the evidence presented supports the claims made throughout the paper.

**Essential References Not Discussed:**

I don’t find essential references that are not discussed.

**Experimental Designs Or Analyses:**

The paper outlines a series of experiments that were conducted to evaluate the effectiveness of the DreamDPO framework in aligning text-to-3D generation with human preferences. Specifically, it mentions two evaluation strategies employed:

1. **Comparison Using a Text-to-Image Reward Model (ImageReward)**: This model was utilized to assess human preferences for the generated 3D assets based on their alignment with provided text prompts. The average preference scores were calculated across 120 rendered images of the 3D assets.
2. **Pairwise Comparisons with GPT-4V**: This strategy involved generating Elo ratings that reflect human judgments on various criteria, including text alignment, 3D plausibility, and texture-geometry coherence. The pairwise comparison results were used to calculate ratings that position the performance of DreamDPO against baseline methods.

The paper does not explicitly state any issues encountered with these experimental designs. However, it does highlight the limitations of prior methods that DreamDPO aims to address, particularly the reliance on accurate pointwise quality evaluations from reward models, which can hinder flexibility and adaptability. Therefore, while the experimental setup leverages innovative measures of evaluation, the potential weaknesses in the dependence on reward models and previous designs may still pose challenges.

Overall, while the paper provides a sound experimental framework, the discussion on limitations suggests that there are ongoing concerns regarding the robustness of the evaluation methods used.

**Methods And Evaluation Criteria:**

The proposed methods and evaluation criteria in the paper make sense for the problem of enhancing text-to-3D generation. The DreamDPO framework introduces an innovative approach that integrates human preferences through a systematic process of direct preference optimization. This method comprises constructing pairwise examples influenced by Gaussian noise, ranking these examples against textual prompts using reward models or large multimodal models, and optimizing the 3D representation.

The evaluation criteria further support this methodology effectively. The paper utilizes two robust evaluation strategies: the ImageReward model, which scores 3D assets based on human preferences by assessing multi-view renderings, and the GPTEval3D benchmark, which compares the generated outputs against 13 baseline methods across five critical criteria—text-asset alignment, 3D plausibility, texture details, geometry details, and texture-geometry coherence.

Both the methodology and the evaluation benchmarks are well-founded and specifically tailored to address the challenges of generating 3D assets that align closely with human expectations, as reflected in the document’s results showing significant improvements in various metrics. Thus, they are appropriate for evaluating text-to-3D generation.

**Other Comments Or Suggestions:**

I don’t have other comments or suggestions.

**Other Strengths And Weaknesses:**

The paper presents the DreamDPO framework, which reflects several notable strengths and weaknesses concerning originality, significance, and clarity.

**Strengths:**

1. **Originality:** The DreamDPO framework introduces a fresh approach to text-to-3D generation by emphasizing direct preference optimization based on human feedback, diverging from traditional methods that rely on pointwise quality evaluations. This optimization-based strategy enhances flexibility and adaptability in generating 3D content, setting it apart from existing techniques that often fail to fully align with human preferences.
2. **Clarity and Structure:** The paper is well-organized, clearly delineating the methodology, experiments, and outcomes. The three-step process of constructing pairwise examples, comparing them, and guiding optimization through preference-driven loss functions is explained clearly, making the complex ideas accessible.
3. **Empirical Validation:** Comprehensive experiments demonstrate that DreamDPO outperforms existing methods, providing robust evidence of its efficacy. This empirical validation strengthens the paper’s claims and offers a solid foundation for the proposed approach.

**Weaknesses:**

1. **Dependence on External Models:** While the framework utilizes reward models or large multimodal models for preference ranking, this dependence may limit the applicability of DreamDPO, especially in scenarios with limited access to such models. The necessity for high-performing models could also restrict its use in real-world applications where computational resources are constrained.
2. **Potential for Bias:** The integration of human preferences into the generation process raises concerns about the inherent biases present in the training data of the reward models. This could lead to outputs that reflect unwanted biases, which may necessitate additional measures to ensure fairness and ethical considerations in generated content.
3. **Time Consuming**: The optimization process takes around two hours on a single NVIDIA RTX A6000 GPU which is too long for text-to-3d generation since some latest method could produce results in less than 1 hour.

**Questions For Authors:**

1. Could you explain how the LMM worked in STEP2 more specific? How to get the questions for LMM? Is it provided by the user or generated from template?
2. Could you explain why you need to use such long time for generation? Is there any methods to speed up the generation process?

**Relation To Broader Scientific Literature:**

The key contributions of the paper build on several important concepts and findings in the broader scientific literature, particularly in the fields of generative models and human preference integration.

1. **Shift from Absolute to Relative Preference Evaluation**: The paper highlights that traditional methods of evaluating 3D generation heavily depend on precise pointwise quality assessments from reward models, which can be restrictive. Previous work has attempted to incorporate human preferences in 3D content generation, but still lacked flexibility and adaptability due to reliance on absolute scoring. DreamDPO shifts this paradigm by utilizing relative preferences, enabling better alignment with human expectations and enhancing the flexibility of the generation process.
2. **Integration of Large Multimodal Models**: The framework proposed in DreamDPO incorporates insights drawn from large multimodal models. This aligns with recent advancements in text-to-image generation, where models have successfully been used to infer and generate content based on textual inputs. By applying such principles to 3D models, DreamDPO takes advantage of the robust features learned from multimodal datasets, thereby establishing a strong connection with ongoing research in multimodal AI.
3. **Direct Preference Optimization**: DreamDPO’s method of optimization through direct preference ranking addresses previous limitations noted in the literature, wherein automated systems struggle to meet diverse user expectations. The incorporation of a preference-driven loss function within the generation process also ties back to reinforcement learning algorithms that have shown efficacy in similar tasks, using human preferences to guide learning in various contexts.

In summary, the contributions of DreamDPO are deeply interwoven with established theories and methods in the literature, pushing the envelope forward by offering a more adaptable and human-aligned approach to text-to-3D generation.

**Theoretical Claims:**

The support of proposed framework appears to be on empirical evaluations and quantitative comparisons rather than theoretical proofs. Thus, there are no specific theoretical claims or proofs outlined that were checked for correctness in the paper. The issues addressed mainly revolve around the limitations of previous methods and how DreamDPO aims to overcome these challenges, rather than validating theoretical aspects through proofs.

---

> ### Author Rebuttal · Authors · 2025-04-01
>
> We thank the reviewer for the positive reviews. We provide our responses below.
>
> > **Q1:** Dependence on External Models.
>
> **A1:** DreamDPO is compatible with external models and rule-based reward metrics, such as image quality metrics (e.g., BRISQUE [1]) and 3D-consistency evaluation. By leveraging 3D geometry information (depth and camera transformation matrices), we could establish stereo correspondence between views to evaluate multi-view consistency. These approaches do not require additional reward models and are suitable for scenarios with limited computational resources. As shown in [Figure 1-1](https://imgur.com/a/XdVx2sn), we include a case study demonstrating DreamDPO with BRISQUE, a no-reference image quality evaluator, to show its effectiveness.
>
> > **Q2:** How to mitigate the potential bias from reward model?
>
> **A2:** Reward hacking is a known issue in RL-based optimization. One solution is to scale the training data and model size of the reward model. As shown in our experiments in Section 4.3, the reward model HPSv2, which has a stronger generalization ability compared to ImageReward, demonstrates superior generation performance in most cases. Moreover, DreamDPO supports an ensemble of reward models, combining their ranking to reduce reliance on a single model and mitigate unwanted biases (see [Figure 1-2](https://imgur.com/a/XdVx2sn)).
>
>
> > **Q3:** How does the LMM in STEP2 work, and where do its questions come from?
>
> **A3:** We detail the LMM-based pairwise comparison in Section B.2. In STEP2, given pairwise examples, LMM conducts the comparison query sequentially. For each query, LMM performs visual question answering based on the provided image and query, and we extract the number of "yes" responses as the score. The questions for LMM can be customized by the user or generated from a template. For instance, LLM can automatically extract questions from a given prompt, such as generating "Is the leaf shouting?" from "A shouting leaf". Alternatively, users can define custom questions, such as "Does the elephant stay on the ground?" for "A dancing elephant".
>
> > **Q4:** Why is the generation time long, and can it be improved?
>
> | Method | MVDream | DreamDPO (10000 Steps) | DreamDPO (4000 Steps) |
> | ------ | ------- | -----------------------| ----------------------|
> | Computation Time | 1 hour | 2 hour | 1.5 hour|
>
> Table 1-1: Reducing generation time for DreamDPO while maintaining improved performance.
>
> **A4:** The vanilla SDS takes around 1 hour for optimization. Our method takes approximately 2 hours due to the pairwise example construction. To speed up the process, we can adopt a simple yet effective strategy: performing SDS for the first 6000 iterations and then switching to DreamDPO. As shown in [Figure 1-3](https://imgur.com/a/XdVx2sn), this approach reduces the generation time by 25% (to around 1.5 hours) while maintaining improved performance.
>
> [1] Anish et al. Blind/referenceless image spatial quality evaluator. ASILOMAR 2011.

---

> > ### Comment · Reviewer_x6sm · 2025-04-05
> >
> > Thank you for the authors’ response to address my concern. I decided to keep my score on this paper.

---

### Decision · Program_Chairs · 2025-05-01

**Decision:**

Accept (poster)

**Comment:**

This paper introduces DreamDPO, an optimization-based framework for text-to-3D generation that aligns outputs with human preferences via Direct Preference Optimization (DPO). The method constructs pairwise examples, ranks them using reward models or large multimodal models (LMMs), and optimizes 3D representations via a preference-driven loss. Experiments on the GPTEval3D benchmark demonstrate superior performance over 13 baselines in terms of text-asset alignment, 3D plausibility, and texture/geometry quality.

DreamDPO presents a novel, empirically validated approach to preference-aligned 3D generation, with clear improvements over reward-based methods (e.g., DreamReward). Most reviewers think that while slower than feed-forward alternatives, its flexibility (LMM/reward-agnostic) and controllability make it a valuable contribution.